# Phenotypic analysis of the unstimulated in vivo HIV CD4 T cell reservoir

Jason Neidleman[1,2†], Xiaoyu Luo[1†], Julie Frouard[1,2], Guorui Xie[1,2], Feng Hsiao[1,2], Tongcui Ma[1,2], Vincent Morcilla[3], Ashley Lee[3], Sushama Telwatte[4], Reuben Thomas[1], Whitney Tamaki[5], Benjamin Wheeler[5], Rebecca Hoh[6], Ma Somsouk[7], Poonam Vohra[8], Jeffrey Milush[5], Katherine Sholtis James[9], Nancie M Archin[9], Peter W Hunt[10], Steven G Deeks[6], Steven A Yukl[4], Sarah Palmer[3], Warner C Greene[1,5], Nadia R Roan[1,2]*

[1]Gladstone Institutes, San Francisco, United States; [2]Department of Urology, University of California, San Francisco, San Francisco, United States; [3]Centre for Virus Research, the Westmead Institute for Medical Research, The University of Sydney, Sydney, Australia; [4]San Francisco Veterans Affairs (VA) Medical Center and University of California, San Francisco, San Francisco, United States; [5]Department of Medicine, University of California, San Francisco, San Francisco, United States; [6]Division of HIV, Infectious Diseases and Global Medicine, University of California, San Francisco, San Francisco, United States; [7]Department of Medicine, Division of Gastroenterology, San Francisco General Hospital and University of California, San Francisco, San Francisco, United States; [8]Department of Pathology, University of California, San Francisco, San Francisco, United States; [9]Division of Infectious Diseases, School of Medicine, University of North Carolina at Chapel Hill, Chapel Hill, United States; [10]Division of Experimental Medicine, University of California, San Francisco, San Francisco, United States

*For correspondence:
nadia.roan@ucsf.edu

†These authors contributed equally to this work

Competing interests: The authors declare that no competing interests exist.

**Abstract** The latent reservoir is a major barrier to HIV cure. As latently infected cells cannot be phenotyped directly, the features of the in vivo reservoir have remained elusive. Here, we describe a method that leverages high-dimensional phenotyping using CyTOF to trace latently infected cells reactivated ex vivo to their original pre-activation states. Our results suggest that, contrary to common assumptions, the reservoir is not randomly distributed among cell subsets, and is remarkably conserved between individuals. However, reservoir composition differs between tissues and blood, as do cells successfully reactivated by different latency reversing agents. By selecting 8–10 of our 39 original CyTOF markers, we were able to isolate highly purified populations of unstimulated in vivo latent cells. These purified populations were highly enriched for replication-competent and intact provirus, transcribed HIV, and displayed clonal expansion. The ability to isolate unstimulated latent cells from infected individuals enables previously impossible studies on HIV persistence.

## Introduction

Combination antiretroviral therapy (ART) suppresses HIV replication but does not eliminate the latent reservoir, which persists for decades in CD4+ T cells and forms a major barrier to HIV cure (*Chun et al., 1997*; *Finzi et al., 1997*; *Wong et al., 1997*). Our understanding of the phenotypic features of latent cells in vivo is limited, in large part because the lack of a universal surface biomarker makes it impossible to directly identify these cells. As a result, fundamental questions remain, such

**eLife digest** There is no cure for the human immunodeficiency virus infection (HIV), but anti-retroviral drugs allow infected people to keep the virus at bay and lead a normal life. These drugs suppress the growth of HIV, but they do not eliminate the virus. If the treatment is interrupted, the virus bounces back within weeks in most individuals. HIV can start growing again because it hides within particular immune cells, called T cells. These infected cells stay in the infected person's body for their whole life in a dormant or "latent" state, and represent the main barrier to an HIV cure. If these cells could be eliminated or prevented from producing more virus without daily treatment, then HIV could be cured. The fact that HIV hides inside T cells has been known for a long time, but it has remained unclear exactly what kinds of T cells the virus prefers.

One challenge to characterizing latently-infected cells is that there is no single protein made by them that is not also made by uninfected T cells. The latently-infected T cells are also very rare: HIV mainly attaches to a protein called CD4, but only one in about a million T cells with CD4 contain the virus. To figure out which CD4-carrying T cells in a patient sample are latently infected, the cells are extracted from the patient's body and 'reactivated' so the virus will start growing again. Unfortunately, the mixture of drugs used to reactivate the T cells changes the cells and the proteins they are producing, which obscures the features the latently-infected T cells had before reactivation.

Neidleman, Luo et al. developed a new approach to trace the infected, reactivated T cells back to their state before reactivation. Using computational methods and a laboratory technique called mass cytometry, the levels of approximately 40 different proteins were measured in millions of T cells from people living with HIV. These experiments provided an 'atlas' of overall T cell features onto which each reactivated cell could be mapped. The population of latently-infected T cells exhibited common features among all the participants. Selecting a few of the most abundant proteins on the surface of the latently-infected cells allowed these cells to be physically separated from all other immune cells.

In the future, this relatively pure population of infected T cells could be used to study how HIV can persist for many decades. The 'map' of these cells' features will provide a valuable resource for HIV researchers and might enable the discovery of new drugs to eliminate the latent T cells.

as whether latent cells distribute equally among all CD4+ T cell subsets, and to what extent the reservoir is similar between individuals and between blood and tissues.

In the absence of direct ways to phenotype latent cells, characterizing the composition of the HIV reservoir has largely entailed sorting CD4+ T cells based on predefined sets of cell-surface proteins, followed by quantitating HIV through a variety of methods. A vast majority of infected cells in virally suppressed individuals harbor defective HIV genomes (*Hiener et al., 2017*; *Ho et al., 2013*; *Imamichi et al., 2016*; *Lee et al., 2017*). Older quantitation methods based on measuring total HIV DNA/RNA levels, regardless of provirus integrity, are being replaced with methods quantitating levels of replication-competent or genome-intact provirus. These methods afford a more accurate picture of the cells leading to viral rebound during treatment interruption, which are the most relevant targets for HIV cure. The 'gold standard' in virus quantitation has been the quantitative viral outgrowth assay (qVOA; *Finzi et al., 1997*). qVOA has not identified significant differences in the amount of replication-competent HIV in the most commonly studied subsets of memory CD4+ T cells – central memory (Tcm), effector memory (Tem), and transitional memory (Ttm) (*Buzon et al., 2014*; *Kwon et al., 2020*). Such observations have led many researchers to consider the replication-competent reservoir as equally residing in all cell subsets, which if true makes targeting it more difficult.

The analysis of proviral sequences, on the other hand, has revealed differences in the frequencies of genome-intact provirus in different sorted subsets. For instance, sequencing of near full-length HIV provirus has identified Th1 and Tem, particularly those that are HLADR+, as subsets preferentially harboring intact viral genomes (*Hiener et al., 2017*; *Horsburgh et al., 2020*; *Lee et al., 2017*). Viral sequencing has also revealed many intact proviruses within clonally expanded populations, particularly within the Tem and Th1 subsets, and those expressing Ox40 or CD161 (*De Scheerder et al., 2019*; *Hiener et al., 2017*; *Kuo et al., 2018*; *Lee et al., 2017*; *Li et al., 2019*). These and

many other studies (*Cohn et al., 2020*; *Liu et al., 2020*) have solidified the view that clonal expansion plays a major role in maintaining the long-lived HIV reservoir.

Although analyses of sorted populations have provided meaningful insights into HIV persistence, they have been limited to comparing cell populations that differ in one or two markers. Thus, they do not provide a comprehensive view of the phenotypic features of latent cells. Several methods have recently enabled the direct phenotyping of reservoir cells at single-cell resolution (*Baxter et al., 2016*; *Cohn et al., 2018*; *Grau-Expósito et al., 2017*; *Pardons et al., 2019*), providing deeper analyses of the reservoir. These methods entail stimulating cells from ART-suppressed patients with a mitogen to induce the expression of HIV RNA and/or proteins, which serve as markers for identifying or isolating cells using FACS. However, one important limitation with these methods is that they require ex vivo stimulation before phenotyping as viral proteins are not typically expressed at detectable levels in unstimulated latent cells (*Baxter et al., 2016*; *Cohn et al., 2018*; *Pardons et al., 2019*). This stimulation will induce phenotypic changes associated with T cell activation, and expression of HIV accessory genes from reactivation will result in host cell remodeling (e.g. downregulation of cell-surface CD4), which will further alter the phenotypes of the cells (*Cavrois et al., 2017*; *Ma et al., 2020*). Therefore, although current methods enable phenotyping of reactivated cells, there is at present no methodology to directly assess the phenotypes of individual latent cells in their original dormant state.

To overcome this limitation, we applied the principles behind Predicted Precursor as determined by SLIDE (PP-SLIDE), an analytic method we recently established that combines high-dimensional single-cell phenotyping and computational approaches. We have used PP-SLIDE to predict the state of CD4+ T cells before phenotypic changes induced by productive HIV infection (*Cavrois et al., 2017*; *Ma et al., 2020*). Here, we applied it to determine the features of in vivo latent CD4+ T cells before their reactivation by ex vivo stimulation, and validated these findings with patient specimens. Through PP-SLIDE analysis, we were able to address questions fundamental to our understanding of HIV latency and to isolate highly enriched populations of replication-competent latent cells for mechanistic studies of reservoir maintenance.

## Results

### Validation of PP-SLIDE as a method to trace reactivated cells back to their latent state

PP-SLIDE was recently validated as an approach to trace HIV-remodeled cells to their original pre-infection states (*Cavrois et al., 2017*; *Ma et al., 2020*). To determine if it can trace reactivated cells to their original latent state, we tested it first in J-Lat cells, an in vitro model of HIV latency. J-Lat cells are Jurkat cells harboring a single integrated HIV provirus that can be reactivated upon stimulation. Multiple lines of J-Lats have been cloned (*Chan et al., 2013*; *Jordan et al., 2003*) that have diverged slightly over successive passages. We tested how well PP-SLIDE traces reactivated cells back to their original clone when given the choice of two parental clones (*Figure 1A*). Cells of clone 6.3 were reactivated by stimulation with PMA/ionomycin and phenotyped using a CyTOF T cell panel (*Ma et al., 2020*), along with unstimulated cells of clone 6.3 and another clone 5A8. Using PP-SLIDE (see Materials and methods), the high-dimensional CyTOF-generated information from reactivated 6.3 cells (identified by expression of HIV Gag) was matched against an atlas of concatenated unstimulated 5A8 and 6.3 cells to identify the most similar unstimulated cell for each reactivated 6.3 cell. These unstimulated cells, identified using a k-nearest neighbor (kNN) approach, are collectively termed 'kNN latent cells'. The error rate, defined as the % of reactivated 6.3 cells that matched to unstimulated 5A8 cells, was 0.8% (*Figure 1B*). These results suggest that although reactivation greatly changes the phenotype of a latent cell, some of the original 'identity' of the cell is still retained and that PP-SLIDE can be used to capture this information and thereby identify precursors of reactivated latent cells.

### Latent cells are not randomly distributed among memory CD4+ T cells in vivo

Next, we applied PP-SLIDE to characterize latent cells from HIV-infected individuals, using a CyTOF panel (*Supplementary file 1*) that is directed at markers of the major subsets and differentiation

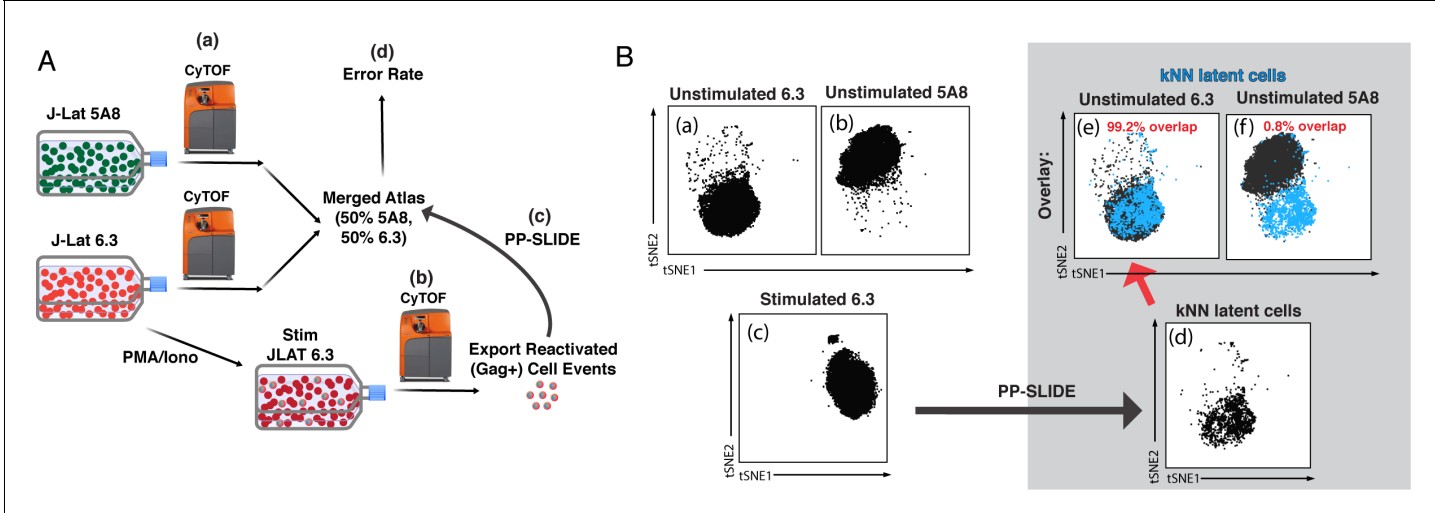

**Figure 1.** Validation of PP-SLIDE as a method to trace reactivated J-Lat clones to their parent. (**A**) Schematic. J-Lat clones 5A8 or 6.3 were phenotyped using CyTOF in the absence of stimulation (a); 6.3 was additionally stimulated for 40 hr with PMA/ionomycin and then phenotyped using CyTOF (b). The high-dimensional datasets of unstimulated 5A8 and 6.3 were merged to generate the 'atlas' of unstimulated J-Lat cells. The high-dimensional data for each reactivated (Gag+) 6.3 cell was matched against the atlas by PP-SLIDE (c). The error rate was defined as the % of reactivated 6.3 cells that matched to unstimulated 5A8 cells (d). (**B**) PP-SLIDE correctly identifies the unstimulated precursor of reactivated 6.3 cells, 99.2% of the time. tSNE plots of unstimulated J-Lat 6.3 (a), unstimulated J-Lat 5A8 (b), and PMA/ionomycin-stimulated J-Lat 6.3 (c). PP-SLIDE was used to identify the kNN latent cell of each reactivated 6.3 cell among a merged atlas of unstimulated 6.3 and 5A8 cells. Note marked overlap of kNN latent cells (d) with unstimulated J-Lat 6.3 (e, 99.2%) and not J-Lat 5A8 (f, 0.8%).

states of CD4+ T cells and includes receptors and intracellular proteins associated with latency (*Banga et al., 2016*; *Buzon et al., 2014*; *Chomont et al., 2009*; *Descours et al., 2017*; *Fromentin et al., 2016*; *Hogan et al., 2018*; *Iglesias-Ussel et al., 2013*; *Khoury et al., 2016*; *Li et al., 2019*; *Serra-Peinado et al., 2019*; *Sun et al., 2015*). Since reactivation events are rare and reducing background was key, we conjugated different anti-Gag antibodies to three different metal lanthanides and only considered cells displaying a Gag signal in all three channels as true events. The full panel was validated in vitro (*Figure 2—figure supplement 1*). We collected via leukapheresis CD4+ T cells from the blood of four ART-suppressed HIV-infected individuals (*Supplementary file 2*), purified memory CD4+ T cells by negative selection, and separated the cells into two pools (*Figure 2A*). One pool was used to generate the atlas of unstimulated cells. This pool was phenotyped within 2 hours of cell isolation, and never went through any cryopreservation or ex vivo culture. The other pool was stimulated with PMA/ionomycin under fully suppressive ART (*Figure 2—figure supplement 2A*) for 40 hr, a time point that precedes mitogen-induced cellular proliferation (*Figure 2—figure supplement 2B*). The CyTOF profiles of reactivated cells were matched against those of the unstimulated atlas cells. Assuming each reactivated cell retained some of its original pre-stimulation identity and that a phenotypically similar cell was present in the atlas of unstimulated cells, we used PP-SLIDE to identify, for each reactivated cell, the nearest-neighbor within the atlas (*Figure 2A*). These identified cells, the 'kNN latent cells', harbor the predicted phenotypes of latent cells before their reactivation from latency.

We identified between 20 and 96 reactivated cells per specimen, corresponding to a mean frequency of 11.3 per million memory (CD45RA-CD45RO+) CD4+ T cells (range 5.6–17.6, *Figure 2B*, *Figure 2—figure supplement 3*). When visualized against the atlas of unstimulated cells by t-distributed stochastic neighbor embedding (tSNE; *van der Maaten, 2009*), Gag+ reactivated cells occupied a unique region, confirming their distinct phenotype relative to unstimulated cells (*Figure 2C*, *left*). kNN latent cells mapped to several distinct regions of the atlas (*Figure 2C*, *right*), suggesting a non-random distribution among memory CD4+ T cells. To verify this, we implemented FlowSOM (*Van Gassen et al., 2015*) to subdivide the atlases from each of the four donors into 20 clusters, and visualized the distribution of kNN latent cells on the tSNE maps or on pie graphs (*Figure 2C*, *center*, and 2D). We found large clusters lacking kNN latent cells as well as small clusters harboring kNN

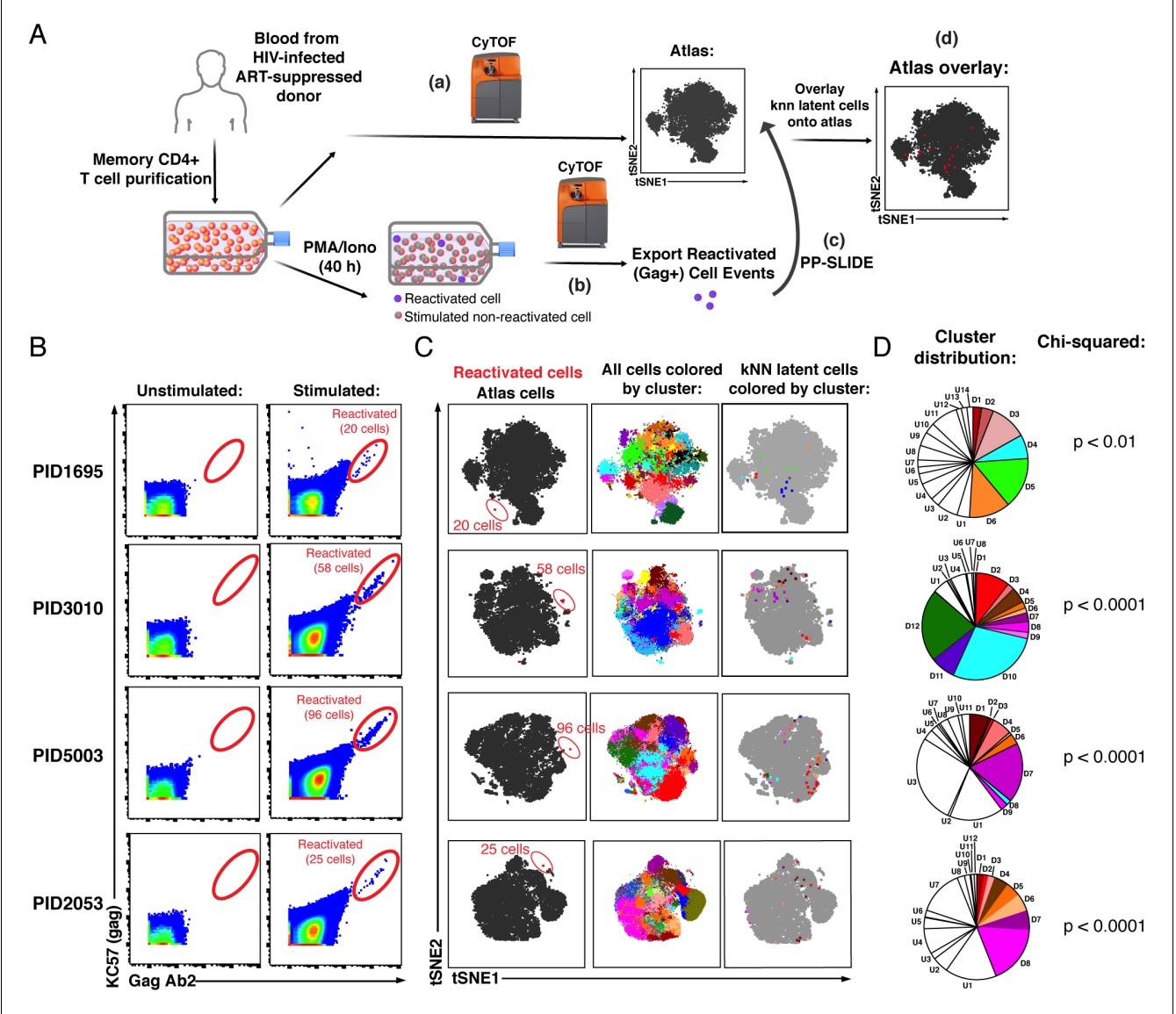

**Figure 2.** PP-SLIDE charting of in vivo latent reservoir reveals non-random distribution of latent cells. (**A**) Schematic of PP-SLIDE analysis of in vivo reservoir. Memory CD4+ T cells from infected, ART-suppressed individuals were isolated and an aliquot immediately processed for CyTOF phenotyping (a), while the remaining cells were phenotyped after treatment with PMA/ionomycin for 40 hr in the presence of ART (b). PP-SLIDE identified a kNN latent cell (the most similar cell in the atlas of unstimulated cells) for each reactivated cell (c). An overlay of the kNN latent cells (red) onto the atlas (gray) reveals regions preferentially associated with latency (d). (**B**) Identification of ex vivo-reactivated latent cells from the blood of ART-suppressed individuals. Cells were phenotyped before or after stimulation for 40 hr with PMA/ionomycin in the presence of ART. Reactivated cells are shown in the gates. The frequencies of reactivated cells were 12.0 (PID1695), 9.9 (PID3010), 17.6 (PID5003), and 5.6 (PID2053) per million memory CD4+ T cells. (**C**) kNN latent cells are non-randomly distributed among memory CD4+ T cells. *Left:* Reactivated cells (red) visualized by tSNE alongside unstimulated memory CD4+ T cells (black) from the same patient. Due to phenotypic changes induced by stimulation and reactivation, the reactivated cells (stacked as tight populations) reside in distinct regions of each tSNE plot (red ovals). *Center:* Atlas of memory CD4+ T cells from each sample, clustered using FlowSOM. Each cluster is depicted in a different color. *Right:* The kNN latent cells are colored according to the cluster they belong to. (**D**) Pie graphs showing relative proportions of each cluster among the atlas. 'D' (Detectable) designates clusters harboring at least one kNN latent cell and 'U' (Undetectable) those lacking any. The D clusters are arranged in order of the frequency of kNN latent cells they harbor, with D1 clusters harboring the highest frequencies. The existence of small D clusters and large U clusters, along with the chi-squared values, demonstrate non-random distribution of the latent reservoir.

The online version of this article includes the following figure supplement(s) for figure 2:

**Figure supplement 1.** CyTOF antibody validation.

**Figure supplement 2.** Establishing conditions for ex vivo reactivation of latent cells.

**Figure supplement 3.** Gating strategy to identify ex vivo reactivated cells from four HIV-infected, ART-suppressed patients.

latent cells, consistent with a non-random distribution of latent cells, which was confirmed using a chi-squared test (*Figure 2D*). For a detailed description of the extent of enrichment of kNN latent cells in clusters and validation of non-random distribution of the reservoir, see Materials and methods.

## kNN latent cells from different donors share phenotypic features

Next, we examined the composition of the kNN latent cells. We focused first on the Tcm, Tem, and Ttm subsets, as they have been the most commonly studied memory CD4+ T cells in the context of HIV latency (*Chomont et al., 2009*; *Josefsson et al., 2013*; *Kwon et al., 2020*; *Pardons et al., 2019*; *Soriano-Sarabia et al., 2014*; *von Stockenstrom et al., 2015*). In all four donors, the contribution of Tem to the reservoir was greater than its contribution to total memory CD4+ T cells (*Figure 3—figure supplement 1*). We did not identify any kNN latent cells with the Ttm phenotype, likely because this subset is relatively rare in blood and not efficiently captured in our samples, each of which harbored fewer than 100 kNN latent cells. When the three FlowSOM clusters of atlas cells with the highest frequencies of kNN latent cells were examined for the relative contributions of the Tcm, Tem, and Ttm subsets, we found that they were all comprised of multiple subsets (*Figure 3—figure supplement 2*). Collectively, these data demonstrate that although Tem are over-represented among kNN latent cells, the clusters most enriched for kNN latent cells – as classified by unbiased clustering based on ~40 markers – are a mixture of Tcm/Tem/Ttm subsets.

These findings could suggest that the reservoir is quite different between individuals. Alternatively, they could mean that latency markers shared between individuals are not those used to define Tcm/Tem/Ttm cells. Consistent with the latter, when datasets from the four donors were combined and re-run within the same tSNE, the kNN latent cells from the four donors resided within a similar region of the plot (*Figure 3A*). These results argue that kNN latent cells between different donors are in fact similar. Consistent with this conclusion, relative to total memory CD4+ T cells, kNN latent cells consistently expressed higher levels of select markers of immune checkpoint (PD1, CTLA4), activation (CD69, CD25, HLADR), and T cell differentiation states (Tbet, CRTH2, CCR6; *Figure 3B*). We conclude that in vivo latent cells are not randomly distributed but instead share specific features between individuals.

Even though latent cells are similar between individuals, the inter-donor variability in latent cells could still be greater than the intra-donor variability. To test this idea, we obtained leukapheresis specimens from donor PID5003 at two time points spaced 2 months apart and compared in an unbiased manner the kNN latent cells from those specimens to kNN latent cells from donor PID3010 (*Figure 3—figure supplement 3A,B*). For each kNN latent cell in the PID3010 sample, we calculated the Euclidean distances (based on CyTOF parameters) to every kNN latent cell in the PID5003 sample. Similar calculations were conducted between kNN latent cells from the two time points sampled in PID5003 (*Figure 3—figure supplement 3A*). The median distances were lower between specimens from the same donor than from different donors (*Figure 3—figure supplement 3C*), demonstrating that despite the similarity of kNN latent cells between donors, the kNN latent cells differ more between donors than within a donor.

## Different ex vivo stimulations reactivate different latent cells

Next, we took advantage of our ability to obtain high-resolution maps of the phenotypic features of latent cells to compare their reactivation by different ex vivo stimulations. Longitudinal blood specimens were obtained from the same suppressed individual and stimulated with the broad-spectrum mitogens PMA/ionomycin or anti-CD3/CD28. Euclidean distance calculations (*Figure 3—figure supplement 3A*) revealed that PMA/ionomycin-reactivatable kNN latent cells from different collection time points were more similar to each other than to anti-CD3/CD28-reactivatable kNN latent cells from the same time point (*Figure 3—figure supplement 3D,E*). Next, we compared the latent cells reactivatable by PMA/ionomycin to those reactivatable by a combination LRA consisting of an HDAC inhibitor together with an ingenol (Romdepsin/PEP005) using three longitudinal blood specimens each spaced 2 months apart. tSNE visualization of the kNN latent cells revealed a region of the plot that harbored kNN latent cells from the PMA/ionomycin samples but not from the Romidepsin/PEP005 samples (*Figure 4A*). Euclidean distance calculations (*Figure 3—figure supplement 3A*) were again used to verify that kNN latent cells reactivatable by the same stimulation at different

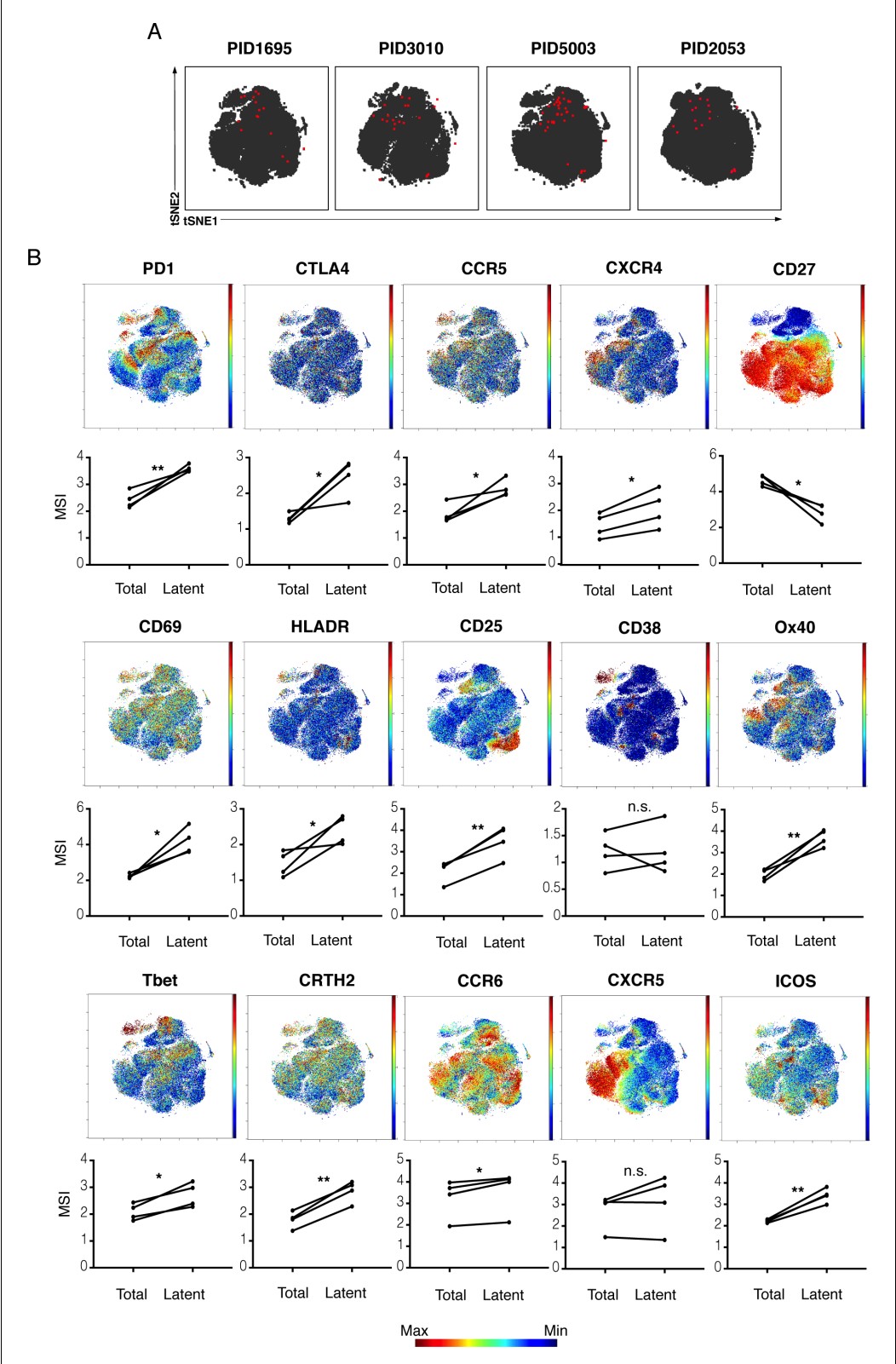

**Figure 3.** kNN latent cells from different donors are phenotypically similar. (**A**) The atlas (*gray*) and kNN latent cells (*red*) for four donors are depicted within the same tSNE space. In all donors, the center region of the tSNE is devoid of kNN latent cells. The kNN latent cells depicted here are those identified in *Figure 2*. (**B**) Expression levels of select antigens on kNN latent cells as compared to the total population of memory CD4+ T cells. Top plots show the antigen expression levels as a heatmap within the tSNE space depicted in *panel A*. Bottom graphs show the mean signal intensity (MSI)

*Figure 3 continued on next page*

*Figure 3 continued*

of each antigen in the total population of memory CD4+ T cells (Total) compared to the kNN latent cells (Latent). Proteins consistently expressed at higher levels on kNN latent cells than on the total population included immune checkpoint molecules (PD1, CTLA4) and the HIV co-receptors CCR5 and CXCR4. Some activation markers (CD69, CD25, HLADR) but not others (CD38) were also upregulated, as were markers of Th1, Th2, and Th17 cells (Tbet, CRTH2, and CCR6, respectively). Some markers of Tfh (PD1, ICOS) but not others (CXCR5) were consistently elevated. By contrast, the Tcm marker CD27 was expressed at low levels on kNN latent cells. *p<0.05, **p<0.01 as determined by a Student's paired t-test. n.s.: not significant.

The online version of this article includes the following figure supplement(s) for figure 3:

**Figure supplement 1.** Distribution of Tcm, Tem, and Ttm among kNN latent cells and total memory CD4+ T cells.
**Figure supplement 2.** The distribution, frequencies, and phenotypic features of clusters most enriched for kNN latent cells.
**Figure supplement 3.** Characterization of the inter-donor and stimulation-dependent variability in kNN latent cells.

time points were more similar than those reactivatable by different stimulations within the same time point (*Figure 4B*). These data suggest that Romdepsin/PEP005 reactivates only a subset of latent cells reactivatable by broad-spectrum mitogens such as PMA/ionomycin, although these findings should be confirmed with longitudinal specimens from additional donors.

## Phenotypic features of tissue kNN latent cells

Since most of the HIV reservoir persists within tissues (*Estes et al., 2017*), we next applied the PP-SLIDE method of latent cell characterization to tissue specimens. One of the four suppressed participants who had donated a leukapheresis specimen, PID3010, also agreed to the collection of two sets of lymph node specimens via fine needle aspirates (FNAs) spaced 6 months apart. Upon isolation, each FNA was immediately mock-treated or stimulated for 40 hr with PMA/ionomycin and then subjected to PP-SLIDE. Unlike with blood specimens, memory CD4+ T cells were not purified ahead of time since FNAs yielded limited numbers of cells. However, we pre-gated on memory CD4+ T cells to compare with purified blood cells. The atlases from the three specimens (one blood, two FNAs) showed that blood and FNA cells were phenotypically distinct (*Figure 4C*). The kNN latent cells from the FNAs, similar to those from blood, were not randomly distributed and remained similar across the longitudinal specimens as demonstrated by their occupying similar regions of tSNE space (*Figure 4C*). Several antigens distinguished kNN latent cells in the blood versus tissues. For instance, CD27, an antigen expressed on Tcm cells, and CD69, a marker of activation in blood but a T resident memory (Trm) marker in tissues (*Cantero-Pérez et al., 2019*), were more highly expressed on FNA kNN latent cells (*Figure 4D*, *Figure 4—figure supplement 1A*). PD1 and CXCR5, markers of Tfh that are preferentially expressed in latent cells particularly in tissues (*Banga et al., 2016*), were also expressed at especially high levels on kNN latent cells from FNAs (*Figure 4D*, *Figure 4—figure supplement 1A*). The co-stimulatory molecule ICOS was expressed at high levels on kNN latent cells exclusively in FNAs, while the exhaustion marker TIGIT was more variably expressed (*Figure 4D*, *Figure 4—figure supplement 1A*). These results reveal marked differences between the phenotypic features of kNN latent cells in blood versus lymph nodes, with the notable exception of PD1, which was high in both compartments.

We also characterized the reservoir in the gut, the primary site of viral persistence during suppressive therapy (*Estes et al., 2017*). As longitudinal gut specimens were not available, we conducted a cross-sectional analysis comparing gut specimens from four donors to the blood specimens previously analyzed by PP-SLIDE. While differences in the reservoir between the blood and gut were observed, some interesting similarities also emerged. kNN latent cells from the gut were mostly CD69$^{hi}$CD27$^{lo}$, while those from blood exhibited a much more variable expression pattern of these two antigens (*Figure 4—figure supplement 1B*). By contrast, kNN latent cells from both the gut and blood expressed high levels of PD1 but relatively low levels of CXCR5 (*Figure 4—figure supplement 1B*). Another similarity was that CTLA4 and CCR6, previously reported to be preferentially expressed on latent cells from tissues (*Gosselin et al., 2017*; *McGary et al., 2017*), were both highly expressed in kNN latent cells from the gut and blood. Taken together with the FNA results, these analyses identify CD69 and PD1 as antigens expressed at high levels in kNN latent cells from both gut and lymph nodes, suggesting shared features of the reservoir between different tissue compartments.

To compare the global similarities between kNN latent cells from the three compartments (blood, lymph nodes, gut), we visualized the atlas and kNN latent cells of all donors combined

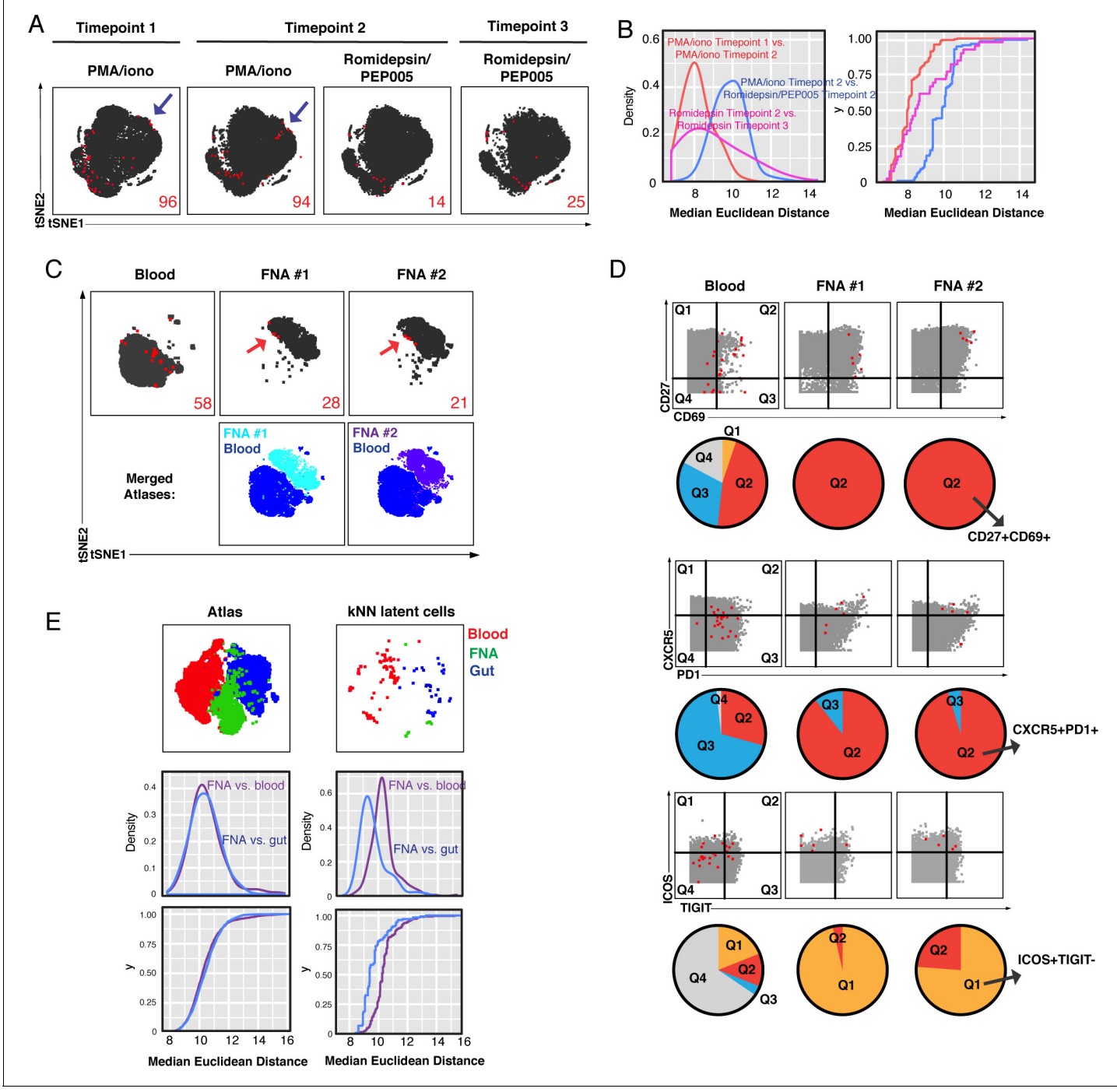

**Figure 4.** Comparison of kNN latent cells reactivatable by different stimulations and from blood versus tissues using longitudinal specimens. (**A**) Atlas (*gray*) and kNN latent cells (*red*) from three blood specimens from the same donor (PID5003) spaced 2–3 months apart, where reactivation was induced by either PMA/ionomycin or Romidepsin/PEP005. Arrows highlight regions that harbor kNN latent cells in the PMA/ionomycin but not Romidepsin/ PEP005 samples. Red numbers indicate the number of kNN latent cells in each plot. (**B**) kNN latent cells reactivatable by PMA/ionomycin at two different time points are more similar to each other than to those reactivatable by Romidepsin/PEP005 at the same time point. The empirical cumulative distribution of median distances between kNN cells was calculated between (1) Time point 1 PMA/ionomycin and Time point 2 PMA/ ionomycin samples (*red*), and (2) Time point 2 Romidepsin/PEP005 and Time point 3 Romidepsin/PEP005 samples (*pink*), and (3) Time point 2 PMA/ ionomycin and Time point 2 Romidepsin/PEP005 samples (*blue*). (**C**) Atlases of memory CD4+ T cells from fine needle aspirates (FNAs) or the blood of the same donor (PID3010) with corresponding kNN latent cells (*red*). Red arrows point to regions of the FNA tSNEs that are concentrated in kNN latent cells. Bottom plots show each FNA atlas overlaid onto the blood atlas. Red numbers indicate the number of kNN latent cells in each plot. (**D**) Dot plots demonstrating that FNA but not blood kNN latent cells (*red dots*) express uniformly high levels of CD27, CD69, CXCR5, PD1, and ICOS. Atlas cells are

*Figure 4 continued on next page*

*Figure 4 continued*

in gray. Pie charts show proportions of kNN latent cells within each quadrant. (E) kNN latent cells from different tissue sites are similar. *Top:* Atlas (*left*) and kNN latent cells (*right*) from four blood, two FNA, and four gut specimens as depicted by tSNE. *Middle/Bottom:* Empirical cumulative distribution of median distances between atlas (*left*) or kNN cells (*right*), as calculated between FNA versus blood (*purple*) or versus gut (*blue*).

The online version of this article includes the following figure supplement(s) for figure 4:

**Figure supplement 1.** Comparison of kNN latent cells from blood and tissues.

using tSNE (*Figure 4E*). Interestingly, the FNA atlas cells appeared equidistant to the blood and gut atlas cells, an observation confirmed by Euclidean distance calculations. By contrast, kNN latent cells from FNA and gut tended to cluster more closely together, away from those from blood, which we also confirmed by Euclidean distance calculations (*Figure 4E*). Together with the above analyses suggesting that kNN latent cells from both tissue compartments express high levels of CD69 and PD1, these results suggest that phenotypic features of the reservoir are shared between different tissue sites. That said, a notable difference was that kNN latent cells from FNAs included both naïve and memory cells, while those from the gut only included memory cells, potentially due to the under-representation of naïve cells within the gut compartment (*Figure 4—figure supplement 1C*).

## Markers of kNN latent cells enrich for the replication-competent, genome-intact reservoir

To determine whether surface markers differentially expressed on kNN latent cells can enrich for the replication-competent reservoir, we custom-designed sorting strategies for each of the four donor blood specimens. Because PD1 is arguably the most validated surface marker of latent cells (*Banga et al., 2016*; *Chomont et al., 2009*; *Fromentin et al., 2016*; *Pardons et al., 2019*), we compared the additional enrichment afforded by our sorting strategy to that afforded by PD1 selection alone. Our panels took into account the expression levels of markers in kNN latent cells, the numbers of cryopreserved cells available from the participant, and the overall abundance of the populations in each sequential gate. The latter two aspects were important considerations as an infrequent population of cells may be highly enriched for kNN latent cells but too rare to sort in sufficient numbers for viral outgrowth and other downstream assays. We arrived at panels of 8–10 markers individualized for each participant. For each donor specimen, we isolated a PD1+ population that we submitted to further selection with the relevant marker panel (enriched) and two PD1+ populations that we did not subject to further selection (disenriched). The disenriched populations harbored markedly lower frequencies of kNN latent cells than the enriched one (*Figure 5*). These populations were sorted (*Figure 6—figure supplement 1*) and subjected to multiple assays.

We first conducted a viral outgrowth assay. Because sorting based on our 8–10 marker panels yielded few cells, we modified the qVOA accordingly. Sorted cells were stimulated in bulk for 2 days with anti-CD3/CD28, and plated into individual wells at various dilutions. The HIV-permissive MOLT-CCR5 cell line was added to propagate viral replication, and after 2 weeks supernatants were quantitated for viral outgrowth by p24 ELISA. Replication was scored as the proportion of wells that had outgrowth and as the concentration of p24 in each well. Both scoring methods revealed strikingly more outgrowth in the enriched relative to both disenriched populations in all four donors (*Figure 6A*). In fact, most disenriched populations did not result in any outgrowth even at the highest concentration tested (200,000 cells/well). The enriched populations also harbored more replication-competent HIV than total memory CD4+ T cells (*Figure 6—figure supplement 2*).

Because recent studies suggest that the in vivo reservoir is transcriptionally active (*Yukl et al., 2018*), we assessed HIV transcription initiation, elongation, and completion in the sorted populations. In the absence of stimulation, HIV transcriptional activity was observed in the sorted cells, with more transcription initiation in the enriched than disenriched ones in 7/8 instances (*Figure 6B*). These results suggest that populations highly enriched for the replication-competent reservoir initiate HIV transcription in the absence of stimulation.

We then assessed to what extent our 8–10 marker panels enrich for latent cells harboring intact genomes. We sequenced by FLIPS (*Hiener et al., 2017*) proviruses from the enriched and from one of the disenriched populations, for each of three donors. Consistent with the PP-SLIDE and outgrowth results, enriched cells harbored higher frequencies of intact p24 sequences than disenriched

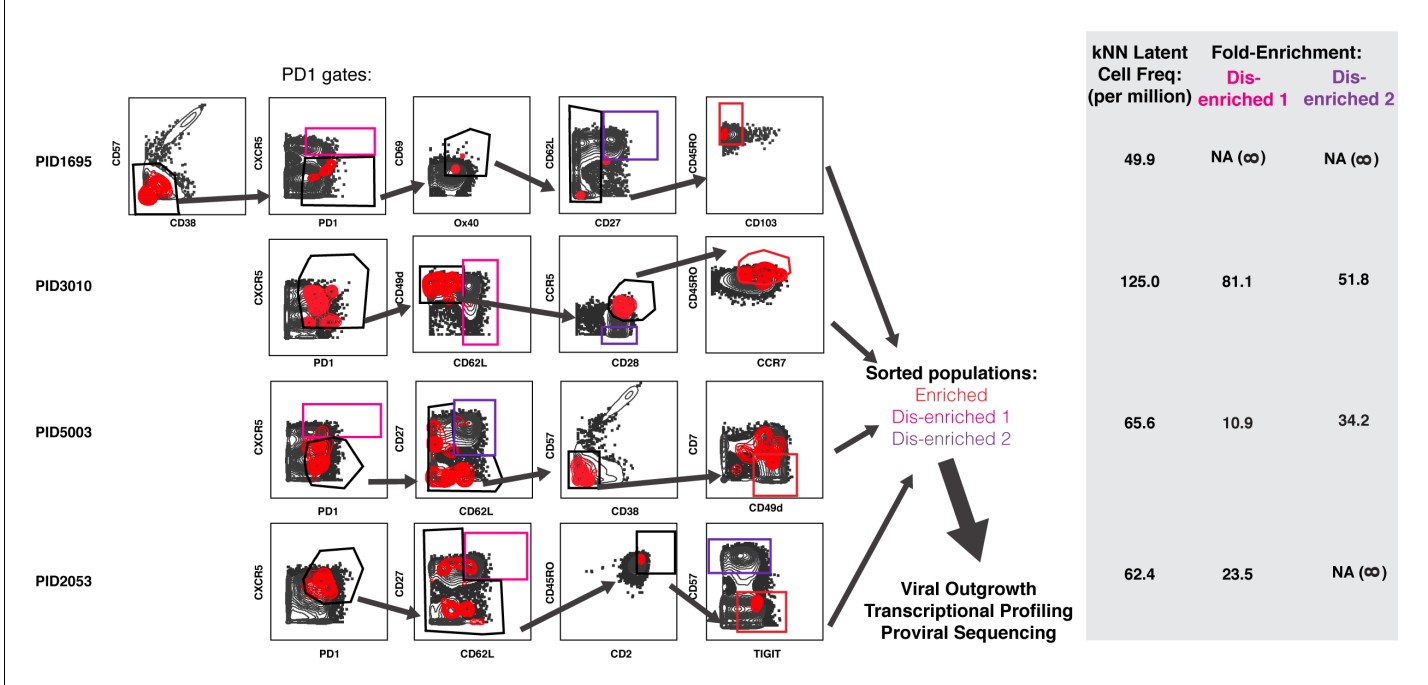

**Figure 5.** Design of tailored sort strategies to isolate three populations memory CD4+ PD1+ T cells harboring different frequencies of kNN latent cells. Sort strategies were designed for each of the four leukapheresis donors analyzed by PP-SLIDE. Shown are the CyTOF datasets, the unstimulated atlas cells are shown as gray contours and kNN latent cells shown as red contours. Gates in black correspond to upstream gates. Each sorting strategy isolates three populations of memory CD4+ PD1+ T cells: two disenriched (*pink, purple*), and one enriched (*red*). Results were pre-gated on live, singlet memory CD4+ T cells (CD3+CD8-CD19-CD45RO+CD45RA-). The three functional assays applied to sorted cells are listed. The gray inset on the right shows frequencies of kNN latent cells in the final enriched populations (per million memory CD4+ T cells) and the fold-enrichment of kNN latent cells in the final sorted enriched population relative to each of the disenriched populations. In instances where a disenriched population did not harbor any kNN cells, the fold-enrichment is listed as NA (not available) because the fold-enrichment is infinity when divided by zero.

ones (*Figure 6C*). To assess the extent of clonal expansion, a major driver of HIV persistence (*Cohn et al., 2020*; *Liu et al., 2020*), we determined the proportions of expanded identical sequences (EIS). EIS were prominent in both populations, but the proportions of EIS with intact p24 sequences were consistently higher in the enriched ones (*Figure 6—figure supplement 3*). A phylogenetic analysis of the sequences revealed that in two of the donors (PID1695 and PID5003), EIS from the enriched and disenriched populations formed distinct lineages, while in donor PID3010 multiple EIS were shared between the enriched and disenriched populations (*Figure 6—figure supplement 4*).

As the enriched population from PID5003 harbored a particularly high frequency of intact p24 sequences, we analyzed this donor in more detail. We categorized all the sequences analyzed from this individual into those that were full-length intact (i.e. no defects along the entire proviral genome), and those harboring large internal deletions, hypermutations, premature stop codons, and inversions. Remarkably, 65.2% of the enriched population was fully intact, compared to only 0.9% for the disenriched population (*Figure 6D*). This corresponded to an overall frequency of 214 full-length intact sequences per million in the enriched population, as compared to only two per million in the disenriched population. The vast majority (95.5%) of the proviruses in the disenriched population harbored large internal deletions. EIS were prominent among both populations (*Figure 6E*). Among the EIS in the enriched population, 66.7% were fully intact, compared to 0% in the disenriched population. Fully-intact EIS in the enriched population comprised two groups, suggesting two major expansions of replication-competent reservoir cells in this population of cells highly enriched for replication-competent HIV (*Figure 6E*). Collectively, these results show that surface markers identified by PP-SLIDE can be used to enrich for replication-competent latent cells and that these cells are enriched for the transcriptionally active reservoir and exhibit clear evidence of clonal expansion of fully-intact proviruses.

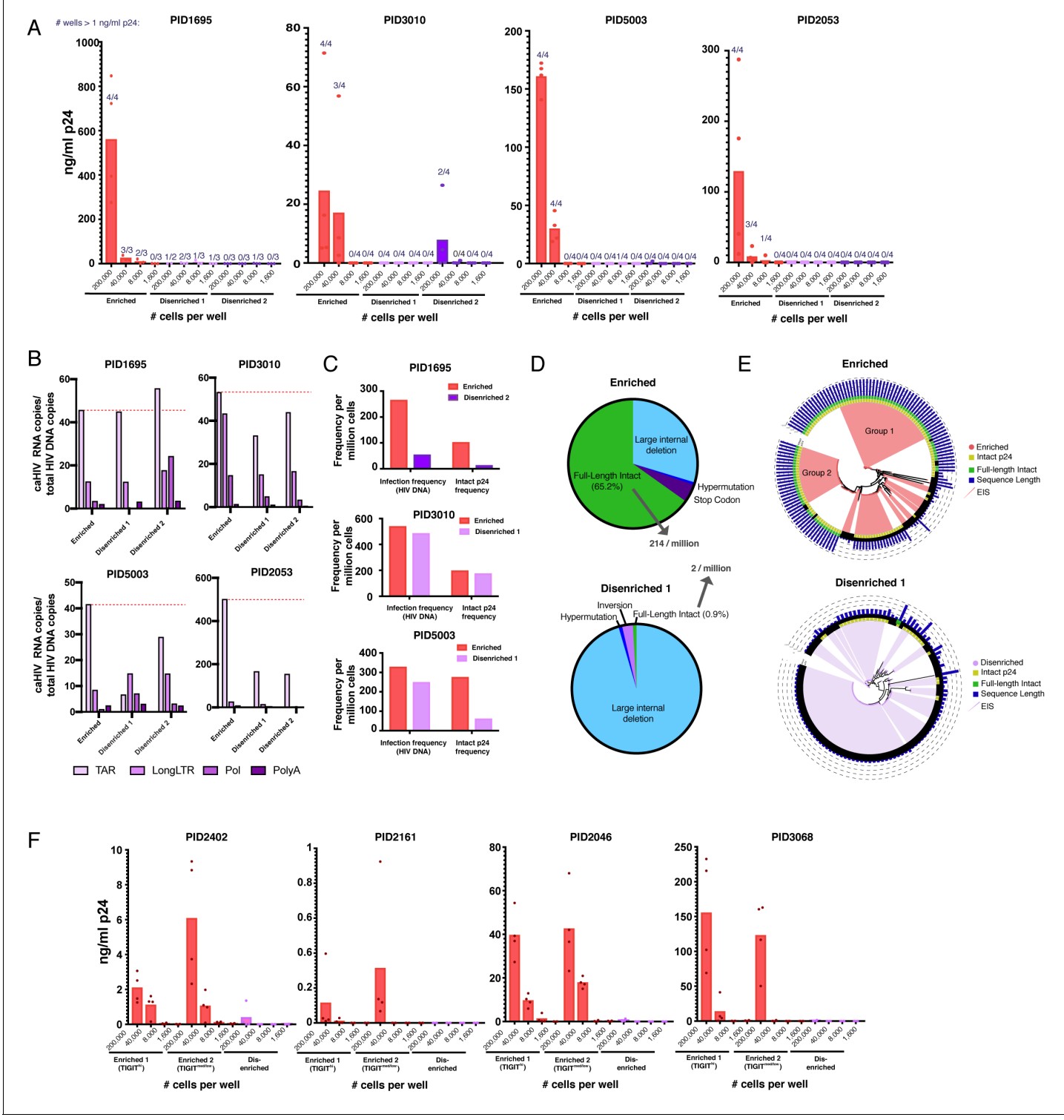

**Figure 6.** Sorting PD1+ cells based on surface markers identified by PP-SLIDE highly enriches for latent cells with replication-competent, transcriptionally active, and clonally expanded HIV. (**A**) Enriched populations of sorted cells are enriched for replication-competent HIV. Three populations of memory CD4+ PD1+ cells – one enriched and two disenriched for kNN latent cells – were purified by sorting (*Figure 6—figure supplement 1*) and then subjected to a viral outgrowth assay. The cells were stimulated for 2 days with anti-CD3/CD28, diluted to the indicated number of cells per well, and cultured with MOLT-CCR5 cells for 15 days. Wells were then assayed for p24 by ELISA. Data are reported as the concentration of p24 in each well (bar graphs), and the fraction of wells that exhibited viral replication as defined by a final p24 content of >1 ng/mL (numbers above each bar). (**B**) Enriched populations harbor more HIV transcription initiation. The enriched and disenriched populations from four

*Figure 6 continued on next page*

*Figure 6 continued*

donors were left unstimulated and quantitated for the levels of initiated (TAR), elongated (Long LTR, Pol), and polyadenylated (PolyA) HIV RNAs. Data are normalized to the corresponding region of HIV DNA. The dotted red line shows the levels of initiated HIV transcripts for the enriched populations. (C) Enriched populations harbored higher frequencies of total HIV and intact p24 sequences in three donors, as revealed by FLIPS. (D) Pie charts showing the proportions of full-length intact and defective proviruses (including large internal deletions, hypermutations, premature stop codons, and inversions) from the enriched (*top*) and disenriched (*bottom*) populations from PID5003. The enriched and disenriched populations harbored full-length intact proviruses at proportions of 66.5% and 0.9%, and at overall frequencies of 214 and 2 per million cells, respectively. (E) Phylogenetic analysis of proviruses demonstrating enrichment of identical full-length intact proviruses in the enriched (*top*) and disenriched (*bottom*) populations from PID5003. Wedges depict expanded identical sequences (EIS). Concentric circles identify proviruses with (*yellow*) or without (*black*) intact p24 sequences and with (*green*) or without (*black*) full-length intact sequences. Blue bars depict the length of each proviral sequence. Groups 1 and 2 labels the two expansions of full-length intact sequences in the enriched population. (F) The universal panel enriches for replication-competent reservoir cells. Memory CD4+ PD1 + from four untested donors were sorted into two enriched (one TIGIT$^{high}$ and one TIGIT$^{med/low}$) and one disenriched population (*Figure 6—figure supplement 6*) and then subjected to the viral outgrowth assay described in *panel A*.

The online version of this article includes the following figure supplement(s) for figure 6:

**Figure supplement 1.** Implementation of tailored sort strategies to isolate three populations of memory CD4+ PD1+ T cells harboring different frequencies of kNN latent cells.
**Figure supplement 2.** Enriched populations of sorted cells harbor more replication-competent HIV than their parent memory CD4+ T cell populations.
**Figure supplement 3.** Relative to disenriched populations, enriched populations harbor more clonally expanded proviruses with intact p24 sequences.
**Figure supplement 4.** Phylogenetic trees of near full-length HIV sequences derived from sorted populations.
**Figure supplement 5.** Design of universal sort panel to enrich for latent cells.
**Figure supplement 6.** Implementation of the universal sort strategy to isolate three populations memory CD4+ PD1+ T cells harboring different frequencies of kNN latent cells from four untested donors.

## Universal panel enriches for latent cells with replication-competent HIV

The sort panels implemented above were tailored for each donor specimen. Given that some features of kNN latent cells are shared between individuals (*Figure 3*), we set out to design a universal panel to enrich for latent cells in a donor-independent fashion. We designed a sorting strategy incorporating eight surface antigens that enrich for kNN latent cells from all four blood specimens analyzed (*Figure 6—figure supplement 5*). As TIGIT was found to be expressed at high levels on kNN latent cells in some but not all donors (*Figure 6—figure supplement 5*), we designed a strategy to isolate two enriched populations – one TIGIT$^{high}$ and one TIGIT$^{med/low}$ – to test the hypothesis that TIGIT is indeed not a marker of latent cells shared between donors. These populations were compared to a PD1+ population disenriched for kNN latent cells (*Figure 6—figure supplement 5*). We applied this 'universal' panel on four donor blood specimens (*Supplementary file 2*) that had not undergone PP-SLIDE analysis (*Figure 6—figure supplement 6*). Viral outgrowth assays revealed robust outgrowth in the enriched but not disenriched populations (*Figure 6F*). Furthermore, the TIGIT$^{high}$ population harbored more viruses in some donors and the TIGIT$^{med/low}$ one in others, validating the notion that TIGIT is not a marker of latent cells shared between donors. These results suggest that shared features between kNN latent cells can be exploited to enrich for the replication-competent latent reservoir across donors.

## Discussion

Although the latent reservoir has long been recognized as a major barrier to HIV cure, the phenotype of these latently infected cells in vivo has remained somewhat obscure. A common viewpoint, based on the limited number of studies that have conducted viral outgrowth assays on sorted cell populations, is that latent HIV persists in all types of CD4+ T cells with no preference for one subset over another. If so, targeting latently infected cells as a curative strategy would be all the more difficult. Taking advantage of recent developments in high-parameter single-cell phenotyping and our PP-SLIDE analysis pipeline, we now show that latently infected cells are not randomly distributed among CD4+ T cells, and provide an in-depth view of the phenotypic features of the in vivo blood and tissue reservoir.

Multiple observations make us confident that PP-SLIDE identifies the physiologically relevant reservoir. First, our in vitro assay with J-Lat cells showed the method could predict the correct precursor cell 99.2% of the time. Second, cells sorted based on PP-SLIDE were enriched for genome-intact and replication-competent reservoir cells in the expected populations. Of note, our PP-SLIDE

characterization had charted the translation-competent reservoir, defined as latent cells able to produce viral proteins upon stimulation (*Baxter et al., 2018*), which may include some replication-incompetent HIV provirus (*Baxter et al., 2016*; *Imamichi et al., 2020*). Nevertheless, our viral outgrowth results closely aligned with the enrichment of kNN latent cells, confirming that PP-SLIDE identifies markers of the replication-competent HIV reservoir most relevant for a cure. Third, using our 'universal panel,' we validated the preferential presence of the replication-competent reservoir in expected populations in four untested donors, and PP-SLIDE's prediction that the latent reservoir is preferentially maintained in TIGIT-expressing cells in some but not all donors.

The sorting strategies used in our validation experiments were designed to enable comparison of different populations of memory CD4+ T cells expressing high levels of PD1, a marker preferentially expressed on reservoir cells (*Banga et al., 2016*; *Chomont et al., 2009*; *Fromentin et al., 2016*; *Pardons et al., 2019*) and which we found expressed at high levels on kNN latent cells. This was implemented for two reasons: First, to demonstrate that we can enrich for latent cells beyond the selection afforded by PD1 alone. Indeed both our viral outgrowth and proviral sequencing results confirmed PP-SLIDE's prediction that some but not all PD1-expressing memory CD4+ T cells harbor high frequencies of latent HIV. Second, to enable mechanistic comparisons of highly similar populations of sorted cells harboring vastly different levels of replication-competent HIV. We found that in 7/8 cases, the enriched populations initiated more HIV transcription than their disenriched counterparts, suggesting that the transcriptionally active reservoir associates most closely with the replication-competent reservoir and therefore should be a target of HIV curative strategies. We also found that relative to disenriched populations, the enriched ones harbored higher proportions of EIS with intact p24, consistent with the importance of clonal expansion in driving the maintenance of the replication-competent HIV reservoir (*Bui et al., 2017*; *Hosmane et al., 2017*; *Lorenzi et al., 2016*), although it should be noted that cells with different integration sites can harbor the same proviral sequence (*Patro et al., 2019*).

Of particular interest was our in-depth comparison of two PD1-expressing memory CD4+ T cell populations from PID5003. A striking 65.2% of proviruses in the enriched population from this donor were fully intact, versus 0.9% in the disenriched, despite both being PD1 expressors. To put these numbers in perspective, on average only 2–3% of sequences from infected individuals treated during chronic infection are intact (*Bruner et al., 2016*; *Hiener et al., 2017*). The overall frequency of intact provirus in the enriched PID5003 population was also extremely high: at 214 per million cells, it was more than an order of magnitude higher than what is observed in Tem (range = 1–26, median, 13 per million), the subset with the highest frequencies of intact provirus as assessed by FLIPS (*Hiener et al., 2017*). This high frequency is not due to an unusually large overall reservoir in this donor, as the phenotypically-similar (PD1+) disenriched population sequenced from this donor exhibited a frequency of only two per million. Our observation that this enriched population, but not the disenriched one, exhibited robust viral outgrowth, further suggests that the intact proviruses in this population are replication-competent. Interestingly, the intact sequences in this enriched population consisted of two large clusters of EIS, suggesting two major clonal expansions in this well-defined population of memory CD4+ cells sorted based on eight surface markers. As these two clusters of viral sequences likely integrated at different genomic sites, these observations support a host rather than virus-driven expansion of these latent cells.

That we could find such vastly different levels of intact and replication-competent reservoir cells in our enriched versus disenriched populations goes against the notion that the reservoir is 'everywhere' and randomly distributed in all subsets. But how do our observations fit with prior reports that the frequencies of replication-competent provirus, as determined from outgrowth assays, were similar between sorted subsets (*Buzon et al., 2014*; *Kwon et al., 2020*)? We believe that the previously sorted subsets were not ones that differ markedly in reservoir frequencies. Most prior subset analyses had focused on the Tem and Tcm subsets, and when focusing on just these subsets our PP-SLIDE results indeed support the notion that both Tem and Tcm were prominent among kNN latent cells. However, our finding that the proportions of Tem were greater among the kNN latent cells than among atlas cells suggests that latency is somewhat biased toward the Tem subset, a finding in line with the higher prevalence of intact provirus in Tem over Tcm (*Hiener et al., 2017*). Further supporting a non-random distribution of the reservoir is our identification of markers consistently expressed at higher levels on kNN latent cells relative to memory CD4+ T cells. Such markers include ones previously implicated in latency (PD1, CTLA4, Ox40) (*Banga et al., 2016*; *Fromentin et al.,*

*2016*; *Kuo et al., 2018*; *McGary et al., 2017*), some activation markers (CD25, CD69), and markers of Th1, Th2, and Th17 cells (Tbet, CRTh2, and CCR6, respectively) suggesting preferential persistence of HIV in more differentiated cells. The reservoir may persist in these cells due to antigen- or homeostatically-driven expansion of these cells which can maintain the reservoir through repeated cycles of clonal expansion.

Our analysis of longitudinal specimens by PP-SLIDE also provided insights into the stability of the reservoir and its capacity for reactivation. By comparing kNN latent cells from different donors versus longitudinal samples, we found intra-individual variability to be less than inter-individual variability. This observation implies that the phenotypic properties of the reservoir are stable over time, a feature that may facilitate its targeting. In fact, longitudinal stability also held for the reservoir's susceptibility to the LRAs we tested. Our observation that different stimulatory signals reactivate different latent cells further suggests that 'shock and kill' strategies may require multiple LRAs, each preferentially targeting different cell subsets, as recently suggested (*Baxter et al., 2016*; *Grau-Expósito et al., 2019*; *Pardons et al., 2019*).

Although our study focused on the blood reservoir for logistical and technical reasons, we also conducted PP-SLIDE analyses on tissue specimens. Perhaps not surprising given the known phenotypic differences between T cells from blood versus tissues (*Wong et al., 2015*), we found that blood and tissue kNN latent cells differ quite dramatically. High-dimensional distance calculations revealed that kNN latent cells were more similar between lymph node and gut than between the tissues and blood, suggesting phenotypic commonalities in the latent reservoir found in tissues. Indeed, kNN latent cells from both lymph node and gut expressed high levels of the Trm marker CD69, suggesting that Trm in these tissues, like those from the cervix (*Cantero-Pérez et al., 2019*), preferentially harbor the reservoir. Together with our observation that kNN latent cells from all sites examined expressed high levels of PD1, these results suggest PD1-expressing Trm as possible targets to eliminate the tissue reservoir. Given recent findings that blood, lymph nodes, and gut are the major sources of HIV dissemination in ART-suppressed individuals (*Chaillon et al., 2020*), targeting latent cells in the compartments analyzed in this study may be sufficient to make a meaningful reduction in the reservoir.

Our study has some limitations, in particular, a relatively small sample size of only eight blood and five tissue donors, although some donors were characterized longitudinally. Nonetheless, our findings do not support the existence of a single surface biomarker that can be used to target the entire HIV reservoir at once. However, PP-SLIDE enables in-depth charting of the latent reservoir residing in the blood and, likely, any tissue we can access. For therapeutic applications, surface antigens identified by PP-SLIDE as preferentially co-expressed on latent cells can be targeted with a series of bi- or multi-specific antibodies (*Klein et al., 2016*). More importantly, our ability to isolate highly enriched populations of replication-competent reservoir cells enables in-depth functional analyses of the reservoir and potentiates future implementation of discovery-based approaches such as scRNAseq and PCR-activated cell sorting (PACS; *Bradley et al., 2018*; *Clark and Abate, 2017*). These approaches hold great promise for identifying novel markers for HIV eradication but require a high frequency of latent cells that was not achievable previously. We envision that implementing PP-SLIDE on cells phenotyped by scRNAseq together with Antibody-seq approaches (*Peterson et al., 2017*; *Stoeckius et al., 2017*) will enable even higher-resolution mapping, with the added advantage that whole-transcriptome approaches do not require pre-defining markers of interest, and allow for the discovery of unanticipated—and perhaps unique—markers of latent cells.

## Materials and methods

### Key resources table

| Reagent type (species) or resource | Designation | Source or reference | Identifiers | Additional information |
|---|---|---|---|---|
| Antibody | HLADR (mouse monoclonal) | Thermofisher | Cat#Q22158 | (1 µg/100 µL) |
| Antibody | RORgt (rat monoclonal) | Fisher Scientific | Cat#5013565 | (1 µg/100 µL) |

*Continued on next page*

*Continued*

| Reagent type (species) or resource | Designation | Source or reference | Identifiers | Additional information |
|---|---|---|---|---|
| Antibody | CD49d (α4) (mouse monoclonal) | Fluidigm | Cat#3141004B | (1 µg/100 µL) |
| Antibody | CTLA4 (mouse monoclonal) | Fisher Scientific | Cat#5012919 | (1 µg/100 µL) |
| Antibody | NFAT (rat monoclonal) | Fluidigm | Cat#3143023A | (1 µg/100 µL) |
| Antibody | CCR5 (mouse monoclonal) | Fluidigm | Cat#3144007A | (1 µg/100 µL) |
| Antibody | BIRC5 (mouse monoclonal) | Fisher Scientific | Cat#MAB886 | (1 µg/100 µL) |
| Antibody | SAMHD1 (rabbit polyclonal) | Proteintech | Cat#12586–1-AP | (1 µg/100 µL) |
| Antibody | Ki67 (mouse monoclonal) | Fisher Scientific | Cat#BDB556003 | (1 µg/100 µL) |
| Antibody | CD95 (mouse monoclonal) | Fisher Scientific | Cat#MAB326100 | (1 µg/100 µL) |
| Antibody | Bcl6 (mouse monoclonal) | Fisher Scientific | Cat#BDB561520 | (1 µg/100 µL) |
| Antibody | CD7 (mouse monoclonal) | Fluidigm | Cat#3147006B | (1 µg/100 µL) |
| Antibody | ICOS (hamster monoclonal) | Fluidigm | Cat#3148019B | (1 µg/100 µL) |
| Antibody | Tbet (mouse monoclonal) | Fisher Scientific | Cat#5013190 | (1 µg/100 µL) |
| Antibody | Gag (71-31) (human monoclonal) | NIH AIDS Reagent | Cat#530 | (1 µg/100 µL) |
| Antibody | Gag (91-5) (human monoclonal) | NIH AIDS Reagent | Cat#1238 | (1 µg/100 µL) |
| Antibody | Gag (241-D) (human monoclonal) | NIH AIDS Reagent | Cat#1244 | (1 µg/100 µL) |
| Antibody | Gag (AG3.0) (mouse monoclonal) | NIH AIDS Reagent | Cat#4121 | (1 µg/100 µL) |
| Antibody | CD2 (mouse monoclonal) | Fluidigm | Cat#3151003B | (1 µg/100 µL) |
| Antibody | CD103 (mouse monoclonal) | Fluidigm | Cat#3151011B | (1 µg/100 µL) |
| Antibody | Gag (28B7) (human monoclonal) | Medimabs | Cat#MM-0289 | (1 µg/100 µL) |
| Antibody | CD62L (mouse monoclonal) | Fluidigm | Cat#3153004B | (1 µg/100 µL) |
| Antibody | TIGIT (mouse monoclonal) | Fludigm | Cat#3154016B | (1 µg/100 µL) |
| Antibody | CCR6 (mouse monoclonal) | BD Biosciences | Cat#559560 | (1 µg/100 µL) |
| Antibody | Gag (KC57) (mouse monoclonal) | Beckman Coulter | Cat#IMBULK1 (custom) | (1 µg/100 µL) |
| Antibody | CD8 (mouse monoclonal) | Biolegend | Cat#301053 | (1 µg/100 µL) |
| Antibody | CD19 (mouse monoclonal) | Biolegend | Cat#302247 | (1 µg/100 µL) |
| Antibody | CD14 (mouse monoclonal) | Biolegend | Cat#301843 | (1 µg/100 µL) |

*Continued on next page*

*Continued*

| Reagent type (species) or resource | Designation | Source or reference | Identifiers | Additional information |
|---|---|---|---|---|
| Antibody | OX40 (mouse monoclonal) | Fluidigm | Cat#3158012B | (1 μg/100 μL) |
| Antibody | CCR7 (mouse monoclonal) | Fluidigm | Cat#3159003A | (1 μg/100 μL) |
| Antibody | CD28 (mouse monoclonal) | Fluidigm | Cat#3160003B | (1 μg/100 μL) |
| Antibody | CD45RO (mouse monoclonal) | Biolegend | Cat#304239 | (1 μg/100 μL) |
| Antibody | CD69 (mouse monoclonal) | Fluidigm | Cat#3162001B | (1 μg/100 μL) |
| Antibody | CRTH2 (rat monoclonal) | Fluidigm | Cat#3163003B | (1 μg/100 μL) |
| Antibody | PD-1 (mouse monoclonal) | Biolegend | Cat#329941 | (1 μg/100 μL) |
| Antibody | CD127 (mouse monoclonal) | Fluidigm | Cat#3165008B | (1 μg/100 μL) |
| Antibody | CXCR5 (rat monoclonal) | BD Biosciences | Cat#552032 | (1 μg/100 μL) |
| Antibody | CD27 (mouse monoclonal) | Fluidigm | Cat#3167006B | (1 μg/100 μL) |
| Antibody | CD30 (mouse monoclonal) | BD Biosciences | Cat#555827 | (1 μg/100 μL) |
| Antibody | CD45RA (mouse monoclonal) | Fluidigm | Cat#3169008B | (1 μg/100 μL) |
| Antibody | CD3 (mouse monoclonal) | Fluidigm | Cat#3170001B | (1 μg/100 μL) |
| Antibody | CD57 (mouse monoclonal) | Biolegend | Cat#359602 | (1 μg/100 μL) |
| Antibody | CD38 (mouse monoclonal) | Fluidigm | Cat#3172007B | (1 μg/100 μL) |
| Antibody | CD4 (mouse monoclonal) | Fluidigm | Cat#3174004B | (1 μg/100 μL) |
| Antibody | CXCR4 (mouse monoclonal) | Fluidigm | Cat#3175001B | (1 μg/100 μL) |
| Antibody | CD25 (mouse monoclonal) | Biolegend | Cat#356102 | (1 μg/100 μL) |
| Antibody | PE-CF594 CD27 (mouse monoclonal) | BD Biosciences | Cat#562297 | (1:25–1:200) |
| Antibody | BUV737 CD38 (mouse monoclonal) | BD Biosciences | Cat#564686 | (1:25–1:200) |
| Antibody | BUV395 CD45RO (mouse monoclonal) | BD Biosciences | Cat#564291 | (1:25–1:200) |
| Antibody | FITC CD57 (mouse monoclonal) | BD Biosciences | Cat#555619 | (1:25–1:200) |
| Antibody | BV650 CD62L (mouse monoclonal) | BD Biosciences | Cat#563808 | (1:25–1:200) |
| Antibody | APC CD69 (mouse monoclonal) | Biolegend | Cat#310910 | (1:25–1:200) |
| Antibody | BV421 CD69 (mouse monoclonal) | BD Biosciences | Cat#562884 | (1:25–1:200) |
| Antibody | APC CD103 (mouse monoclonal) | BD Biosciences | Cat#563883 | (1:25–1:200) |

*Continued*

| Reagent type (species) or resource | Designation | Source or reference | Identifiers | Additional information |
|---|---|---|---|---|
| Antibody | APC-R700 CXCR5 (CD185) (rat monoclonal) | BD Biosciences | Cat#565191 | (1:25–1:200) |
| Antibody | PE-Cy7 CD134 (OX40) (mouse monoclonal) | BD Biosciences | Cat#563663 | (1:25–1:200) |
| Antibody | PE CD279 (PD-1) (mouse monoclonal) | BD Biosciences | Cat#560795 | (1:25–1:200) |
| Antibody | BUV737 CD28 (mouse monoclonal) | BD Biosciences | Cat#612815 | (1:25–1:200) |
| Antibody | BB515 CD49d (mouse monoclonal) | BD Biosciences | Cat#564593 | (1:25–1:200) |
| Antibody | PE/Dazzle 594 CD197 (CCR7) (mouse monoclonal) | Biolegend | Cat#353236 | (1:25–1:200) |
| Antibody | Brilliant Violet 421 CD195 (CCR5) (mouse monoclonal) | Biolegend | Cat#359118 | (1:25–1:200) |
| Antibody | PE-Cy7 CD57 (mouse monoclonal) | Thermofisher | Cat#25-0577-42 | (1:25–1:200) |
| Antibody | BUV395 CD25 (mouse monoclonal) | BD Biosciences | Cat#564034 | (1:25–1:200) |
| Antibody | APC-H7 CD7 (mouse monoclonal) | BD Biosciences | Cat#564020 | (1:25–1:200) |
| Antibody | BV605 CD127 (mouse monoclonal) | BD Biosciences | Cat#562662 | (1:25–1:200) |
| Antibody | APC TIGIT (mouse monoclonal) | Biolegend | Cat#372706 | (1:25–1:200) |
| Antibody | PE-Cy5 CD28 (mouse monoclonal) | Biolegend | Cat#302910 | (1:25–1:200) |
| Antibody | APC-Cy7 CD69 (mouse monoclonal) | BD Biosciences | Cat#560737 | (1:25–1:200) |
| Antibody | BV421 CD2 (mouse monoclonal) | Biolegend | Cat#309218 | (1:25–1:200) |
| Antibody | PE-Cy7 CD57 (mouse monoclonal) | Thermofisher | Cat#25-0577-42 | (1:25–1:200) |

## Study participants and specimen collection

Multiple leukapheresis samples were obtained from eight HIV-1 subtype-B individuals on long-term ART who initiated therapy during chronic (>1 year) infection and were stably suppressed (HIV RNA <40 copies/mL; *Supplementary file 2*). Leukapheresis specimens from four of these eight individuals were subjected to full analyses using CyTOF/PP-SLIDE. Lymph node specimens were obtained from one of these four participants under local anesthesia using ultrasound-guided fine needle aspiration. All fine needle aspirate (FNA) procedures were performed without any adverse effects, and the subject reported minimal to no discomfort both during and after the procedure and was able to perform their usual activities after the procedure. Gut specimens were obtained by sigmoid biopsies from four ART-suppressed HIV-infected participants (*Supplementary file 2*). All participants provided informed consent before participation. The study was approved by the University of California, San Francisco (IRB # 10–01330) and the University of North Carolina (IRB # 12–1660).

## Cells

### J-Lat culture and stimulation

J-Lat clone 5A8 was previously described (*Chan et al., 2013*) and J-LAT clone 6.3 was obtained from the NIH AIDS Reagent Program. The identities of both cell lines were authenticated and tested for mycoplasma contamination within the past 10 years. Both cell lines were cultured in RP10 media (RPMI 1640 medium [Corning] supplemented with 10% fetal bovine serum [FBS, VWR], 1% penicillin

[Gibco], and 1% streptomycin [Gibco]). Cells were either processed for CyTOF in the absence of stimulation or stimulated with 16 nM phorbol myristate acetate (PMA, Sigma-Aldrich) and 0.5 µM ionomycin (Sigma-Aldrich) for 40 hr before processing for CyTOF analysis as described in the appropriate section below.

## Preparation of HLACs
Uninfected and HIV-infected human lymphoid aggregate cultures (HLACs) were used as a source of cells to validate the CyTOF panel. Human tonsils from the Cooperative Human Tissue Network (CHTN) were dissected into small pieces and pressed through a 40 µm strainer (Fisher). The resulting HLAC cells were cultured at a concentration of $10^6$ cells/well in 96-well U-bottom polystyrene plates (VWR) in Complete Media (CM) (consisting of RPMI 1640 supplemented with 15% FBS, 100 µg/mL gentamicin [Gibco], 200 µg/mL ampicillin [Sigma-Aldrich], 1 mM sodium pyruvate [Sigma-Aldrich], 1% non-essential amino acids [Mediatech], 1% Glutamax [ThermoFisher], and 1% Fungizone [Invitrogen]) for 3 days in the absence or presence of 100–200 ng/mL p24$^{Gag}$ of F4.HSA. Both uninfected and infected cells were then treated with cisplatin and PFA as described in the appropriate section below and analyzed using CyTOF.

## Memory CD4+ T cell enrichment of PBMCs
PBMCs were isolated from leukapheresis specimens using Lymphoprep (StemCell Technologies). A portion of the cells was immediately cryopreserved by pelleting the cells and resuspending them at a concentration of $50-100 \times 10^6$ cells/mL in RPMI 1640 medium (Corning) with 90% FBS and 10% DMSO (Sigma-Aldrich). CD4+ T cells were then purified by negative selection using the EasySep Human CD4+ T Cell Enrichment Kit (Stemcell Technologies), and then further enriched for memory cells by negative selection with CD45RA MicroBeads (Miltenyi Biotec) to remove naïve cells.

## Isolation and preparation of lymph node cells
Lymph node specimens were obtained by FNA biopsies of 1–2 lymph nodes in the inguinal chain area. For sampling, 3cc slip tip syringes and either 23- or 22-gauge needles, 1 ½ inch in length, were used. Generally, 4–5 passes per lymph node were performed. A portion of the first pass was taken for fixation in 95% alcohol onto a slide, which was then stained with Toluidine Blue and then examined by microscopy. After confirmation from the cytopathologist that the sample was representative of the target lymph node, the remaining sample in the needle was transferred into RPMI medium (Corning). Due to the small numbers of cells from these specimens, the cells were neither cryopreserved nor enriched for CD4+ T cells and were immediately prepared for PP-SLIDE analysis. A total of two FNAs were drawn from the same participant (PID3010) ~6 months apart.

## Isolation and preparation of gut cells
Processing of sigmoid biopsies into single-cell suspensions was performed similar to approaches recently described (*Trapecar et al., 2017*). Briefly, tissue specimens were first incubated for 20 min at 37°C with rotation in intraepithelial lymphocyte (IEL) solution (consisting of 200 mL DPBS, 10 mM DTT [Sigma-Aldrich], 5 mM EDTA [Thermofisher], 10 mM HEPES [Thermofisher], and 5% FBS), after which supernatants were saved and tissues subjected to two additional rounds of treatment. Media from these incubations were combined, filtered through a 70 micron filter (Fisher), and the resulting IELs were resuspended in CM. The tissue pieces remaining after the IEL incubations were incubated for 20 min at 37°C with 10 mL rinse buffer (RPMI 1640 with 10 mM HEPES and 5% FBS). Tissue pieces were then placed into a gentleMACS C Tube (Miltenyi Biotec) in 1 mL of digestion solution consisting of 6 mL RPMI-1640, 10 mM HEPES, 5% FBS, 6 mg collagenase (Worthington-Biochemical Corp), and 7.5 µg/mL DNAse (Sigma-Aldrich), and rotated for 30 min at 37°C. About 2 mL of Rinse Buffer was then added for a total of 3 mL. The sample was vortexed for 10 s and the sample processed through the GentleMACS Dissociator (Miltenyi Biotec) to isolate cells from the lamina propria. The tissue specimens were then mechanically dissociated for 10 passes using a 5 mL syringe and a blunt, 20G needle (Becton Dickinson). After adding 5 mL Rinse Buffer, the sample was passed through a 70-micron strainer, washed with another 10 mL Rinse Buffer, and passed through a new 70-micron strainer. The flow-through material containing cells was then pelleted and washed once with 5 mL Rinse Buffer. The cells were then treated at 4°C for 30 min with 1 mL of DNAse solution (6

mL RPMI-1640, 10 mM HEPES, 5% FBS, and 7.5 µg/mL DNAse (Sigma-Aldrich)). The cells were then washed once in CM, and then combined with the IELs. Of note, these gentle processing conditions were established and optimized to minimize digestion-induced changes in surface antigen expression (*Trapecar et al., 2017*). Due to the small number of cells from these specimens, the cells were neither cryopreserved nor enriched for CD4+ T cells, and were immediately prepared for PP-SLIDE analysis. In one of the donors, half of the cells were used in a viral outgrowth assay.

## Kinetics of proliferation following mitogen stimulation

Memory CD4+ T cells from the blood of two HIV-seronegative donors were purified as described above, and then resuspended at $2 \times 10^7$ cells/mL in PBS containing 0.1% FBS. The cells were then loaded for 8 min with 1.5 µM of the proliferation dye CFSE (ThermoFisher). Labeling was then stopped by the addition of an equal volume of pre-warmed, 100% FBS, and the labeled cells were incubated at 37°C for an additional 10 min. The sample was then washed three times in RP10. Memory CD4+ T cells not exposed to CFSE were treated identically in parallel. The time = 0 specimen was immediately stained as described below. The remaining cells were cultured with or without activation with 16 nM PMA, 1 µM ionomycin, and 100 IU/mL IL2 (all three reagents from Thermofisher). At the indicated time points, cells were stained with APC/Cyanine7 anti-human CD3 (Clone SK7, Biolegend), PE/Cy7 anti-human CD4 (Clone A161A1, Biolegend), Alexa Fluor 700 anti-human CD8 (Clone SK1, Biolegend), and Zombie Red (Biolegend) as a Live/Dead discriminator. Stained cells were fixed and analyzed by FACS on an LSRII (BD Biosciences). Flowjo (BD Biosciences) was used for analysis. Live, singlet CD3+CD4+CD8- cells were assessed for proliferation by monitoring the loss of CFSE signal. Results shown are from one of two donors which gave similar results.

## Viral production

293 T cells were seeded in 6-well plates (Falcon) at a concentration of $3 \times 10^5$ cells/well, and transfected the next day using FuGENE (Promega) with 0.5 µg/well of the F4.GFP or F4.HSA proviral constructs previously described (*Cavrois et al., 2017*; *Ma et al., 2020*; *Neidleman et al., 2017*). Supernatants from the cultures were harvested after 2 days and p24[Gag] concentrations were measured with a Lenti-X p24 Rapid Titer kit (Clontech).

## In vitro confirmation of the ART cocktail's suppressive activity

To demonstrate that the cocktail of ART used in this study was fully suppressive, PBMC-derived CD4 + T cells from HIV-seronegative donors were activated for 2 days with RP10 containing 10 µg PHA (PeproTech) and 100 IU/mL IL2 (Thermofisher), washed with RP10, and then cultured in 20 IU/mL IL2 for one additional day. The cells were then either left untreated, or pretreated for 1 hr with an ART cocktail consisting of 5 µM AZT, 5 µM Ritonavir, 8 nM Efavirenz, 10 µM Lamivudine, 50 nM Raltegravir, and 0.5 µg/mL T-20 (all from NIH AIDS reagent program). Cells were then cultured for 4 days at a concentration of $2 \times 10^6$ cells/mL with RP10 alone or in the presence of 250 ng/mL p24[Gag] F4.GFP in RP10 containing 20 IU/mL IL2. The cells were then stained for 30 min at 4°C at a concentration of $10^6$ cells/well in FACS buffer, with anti-CD3 APC/Cyanine7, anti-CD4 PE/Cy7, anti-CD8 Alexa Fluor 700, and Zombie Aqua as a Live/Dead discriminator (all from Biolegend). The cells were then washed three times with FACS buffer and fixed overnight in 2% PFA (Electron Microscopy Sciences). The cells were analyzed on an LSRII (BD Biosciences) and Flowjo software was used to gate on live, singlet CD3+CD8- cells and infection rates in these cells were monitored by GFP expression.

## Preparation of participant specimens for PP-SLIDE analysis

For atlas generation, we used freshly isolated and purified memory CD4+ T cells from blood, or freshly isolated total cells from tissues (as described above). A portion of these cells was immediately treated with cisplatin (Sigma-Aldrich) as a Live/Dead marker and fixed. Briefly, $6 \times 10^6$ cells were resuspended at room temperature in 2 mL PBS (Rockland) with 2 mM EDTA (Corning). Next, 2 mL of PBS containing 2 mM EDTA and 25 µM cisplatin (Sigma-Aldrich) were added to the cells. The cells were quickly mixed and incubated at room temperature for 60 s, after which 10 mL of CyFACS (metal contaminant-free PBS [Rockland] supplemented with 0.1% FBS and 0.1% sodium azide [Sigma-Aldrich]) was added to quench the reaction. The cells were then centrifuged and resuspended in 2% PFA in CyFACS and incubated for 10 min at room temperature. The cells were then

washed twice in CyFACS, after which they were resuspended in 100 μL of CyFACS containing 10% DMSO. These fixed cells were stored at −80℃ until analysis by CyTOF.

The rest of the freshly isolated cells were stimulated to allow for latent cell reactivation. These cells were diluted to $2 \times 10^6$ cells/mL and stimulated with 16 nM PMA (Sigma-Aldrich) and 1 μM ionomycin (Sigma-Aldrich) for 40 hr. This stimulation time was chosen as it is similar to times previously implemented (*Cohn et al., 2018*) and shorter stimulations under these conditions did not lead to reactivation. Where indicated, cells were instead stimulated with Dynabeads Human T-Activator CD3/CD38 beads (Gibco), or 12 nM PEP005 (Ingenol 3-angelate, Sigma-Aldrich) with 40 nM romidepsin (Sigma-Aldrich). For gut specimens, to minimize cellular toxicity, cells were instead stimulated for 16 hr with 160 nM PMA and 1 μM ionomycin. All stimulations were conducted in the presence of 100 IU/mL IL2 (Thermofisher). To prevent spreading infection, a cocktail of ART consisting of 5 μM AZT, 5 μM Ritonavir, 8 nM Efavirenz, 10 μM Lamivudine, 50 nM Raltegravir, and 0.5 μg/mL T-20 was added. To limit the death of reactivated cells, 10 μM of the pan-caspase inhibitor Z-VAD-FMK (R and D Systems Inc) was added as previously described (*Cohn et al., 2018*; *Grau-Expósito et al., 2019*).

## CyTOF staining and data acquisition

Staining of cells for analysis by CyTOF was conducted similar to recently described methods (*Cavrois et al., 2017*; *Ma et al., 2020*; *Trapecar et al., 2017*). Briefly, cisplatin-treated cells were thawed and washed in Nunc 96 DeepWell polystyrene plates (Thermo Fisher) with CyFACS buffer at a concentration of $6 \times 10^6$ cells/800 μL per well. Cells were then pelleted and blocked with mouse (Thermo Fisher), rat (Thermo Fisher), and human AB (Sigma-Aldrich) sera for 15 min at 4℃. The samples were then washed twice in CyFACS, pelleted, and stained in a 100 μL cocktail of surface antibodies (*Supplementary file 1*) for 45 min at 4℃. The samples were then washed 3× with CyFACS and fixed overnight at 4℃ in 100 μL of freshly prepared 2% PFA in PBS (Rockland). Samples were then washed twice with Intracellular Fixation and Permeabilization Buffer (eBioscience) and incubated in this buffer for 45 min at 4℃. Next, samples were washed twice in Permeabilization Buffer (eBioscience). The samples were then blocked for 15 min at 4℃ in 100 μL of mouse and rat sera diluted in Permeabilization Buffer, washed 1× with Permeabilization buffer, and incubated for 45 min at 4℃ in a 100 μL cocktail of intracellular antibodies (*Supplementary file 1*) diluted in Permeabilization Buffer. The cells were then washed with CyFACS and stained for 20 min at room temperature with 250 nM of Cell-ID Intercalator-IR (Fluidigm). Finally, the cells were washed 2× with CyFACS buffer, once with MaxPar cell staining buffer (Fluidigm), and once with Cell acquisition solution (CAS, Fluidigm), and then resuspended in EQ Four Element Calibration Beads (Fluidigm) diluted to 1× in CAS. The sample concentration was adjusted to acquire at a rate of 200–350 events/sec using a wide-bore (WB) injector on a CyTOF2 instrument (Fluidigm) at the UCSF Parnassus flow core facility.

For tissue specimens where multiple specimens were combined to minimize cell loss, the Cell-ID 20-Plex Pd Barcoding Kit (Fluidigm) was used according to the manufacturer's instructions to barcode the individual cisplatin-treated samples before mixing. Briefly, the frozen and fixed samples were thawed and washed once with CyFACS and 2× with Maxpar Barcode Perm Buffer (Fluidigm), and then resuspended in 800 μL of Barcode Perm Buffer. A total of 10 μL of each barcode was diluted in 100 μL of Barcode Permeabilization Buffer and added to each sample. The samples were barcoded for 30 min at room temperature, washed once with MaxPar cell staining buffer, and once with CyFACS. The samples were then combined subjected to the CyTOF staining protocol as described above.

## Cell sorting

Memory CD4+ T cells purified from cryopreserved PBMCs as described above were sorted using the following antibodies. Antibody clone numbers are indicated in the table further below:

Panel #1 (used for PID1695):
CD27 PE-CF594; CD38 BUV737; CD45RO BUV395; CD57 FITC; CD62L BV650; CD69 BV421; CXCR5 APC-R7; OX40 PE-Cy7; PD-1 PE
Panel #2 (used for PID3010):

CD45RO BUV395; CD62L BV650; CD69 APC; CXCR5 APC-R700; OX40 PE-Cy7; PD-1 PE; CD28 BUV737; CCR5 BV421; CD49d BB515; CCR7 PE-Dazzle

Panel #3 (used for PID5003):
CD27 PE-CF594; CD62L BV650; CD57 PE-Cy7; CD38 BUV737; CD25 BUV395; CCR5 BV421; CD49d BB515; CD7 APC Cy7; CD127 BV 605; TIGIT APC; PD-1 PE; CXCR5 APC R7; CD28 PE-Cy5

Panel #4 (used for PID2053):
CXCR5 APC R7; PD-1 PE; CD27 PE-CF594; CD62L BV650; CD45RO BUV395; CD2 BV421; CD57 FITC; TIGIT APC; CD69 APC-Cy7; OX40 PE-Cy7

Panel #5 (universal panel used for untested donors PIDs 2402, 2781, 2161, and 2046):
PD-1 PE; CD49d BB515; CD57 PE-Cy7; CD38 BUV737; CD27 PE-CF594; CD62L BV650; CD45RO BUV395; CD2 BV421; TIGIT APC

Dead cells were excluded using the Zombie Aqua fixable viability kit (BioLegend), while doublets were excluded based on FSC-A/FSC-H. Cells were sorted on an Aria FACS sorter (Becton Dickinson). The purity of the subsets was confirmed by analysis on the Aria immediately post-sort.

## Sorting antibodies

| Antigen target | Clone | Vendor (catalog number) |
|---|---|---|
| Live/Dead (Zombie Aqua) | | BioLegend (#423102) |
| PE-CF594 Mouse Anti-Human CD27 | M-T271 | BD (#562297) |
| BUV737 Mouse Anti-Human CD38 | HB7 | BD (#564686) |
| BUV395 Mouse Anti-Human CD45RO | UCHL1 | BD (#564291) |
| FITC Mouse Anti-Human CD57 | NK-1 | BD (#555619) |
| BV650 Mouse Anti-Human CD62L | DREG-56 | BD (#563808) |
| APC Mouse Anti-Human CD69 | FN50 | BioLegend (#310910) |
| BV421 Mouse Anti-Human CD69 | FN50 | BD (#562884) |
| APC Mouse Anti-Human CD103 | Ber-ACT8 | BD (#563883) |
| APC-R700 Rat Anti-Human CXCR5 (CD185) | RF8B2 | BD (#565191) |
| PE-Cy7 Mouse Anti-Human CD134 (OX40) | ACT35 | BD (#563663) |
| PE Mouse anti-Human CD279 (PD-1) | EH12.1 | BD (#560795) |
| BUV737 Mouse Anti-Human CD28 | CD28.2 | BD (#612815) |
| BB515 Mouse Anti-Human CD49d | 9F10 | BD (#564593) |
| PE/Dazzle 594 Mouse Anti-Human CD197 (CCR7) | G043H7 | BioLegend (#353236) |
| Brilliant Violet 421 Rat Anti-Human CD195 (CCR5) | J418F1 | BioLegend (#359118) |
| PE-Cy7 Mouse Anti-Human CD57 | TB01 | eBioscience (25-0577-42) |
| BUV395 Mouse Anti-Human CD25 | 2A3 | BD (#564034) |
| APC-H7 Mouse Anti-Human CD7 | M-T701 | BD (#564020) |
| BV605 Mouse Anti-Human CD127 | HIL-7R-M21 | BD (#562662) |
| APC Mouse Anti-Human TIGIT | A15153G | BioLegend (#372706) |
| PE-Cy5 Mouse Anti-Human CD28 | CD28.2 | BioLegend (#302910) |
| APC-Cy7 Mouse Anti-Human CD69 | FN50 | BD (#560737) |
| BV421 Mouse Anti-Human CD2 | TS1/8 | BioLegend (#309218) |
| PE-Cy7 Mouse Anti-Human CD57 | TBO1 | Invitrogen (#25-0577-42) |

## Viral outgrowth assays

Cells sorted from blood were cultured at a density of $10^6$ cells/mL in RPMI in the presence of 60 IU/mL recombinant human IL-2 (Thermofisher), and activated with Dynabeads Human T-Activator CD3/

CD28 (Gibco) at a ratio of 1 bead/cell. Of note, all populations of cells were sorted before stimulation to account for any effects of the sorting process on viral outgrowth. After 48 hr of culture, cells were diluted to a concentration of $10^5$ cells/mL in RPMI supplemented with 60 IU/mL recombinant human IL-2, and then plated at $2 \times 10^5$ cells per well (dilution A), $4 \times 10^4$ cells per well (dilution B), $8 \times 10^3$ cells per well (dilution C), and $1.6 \times 10^3$ cells per well (dilution D). Molt4/CCR5 cells (*Laird et al., 2013*) were then added at $10^5$ cells per well. Cultures were replenished at days 3, 6, 12, and 15 by replacing half of the media with RPMI containing 5 IU/mL IL-2 without disturbing the cellular layer. On day 9, half the cultures, inclusive of both cells and media, were discarded, and the remaining cells were replenished with the same volume of RPMI containing $10^5$ Molt4/CCR5 cells. At the end of the culture period (days 15–18), supernatants were quantitated for levels of p24 antigen by ELISA (Takara).

A viral outgrowth assay was conducted for one of the gut specimens (PID1223) to demonstrate the presence of replication-competent virus in the tissue. To this end, a Digital ELISA Viral Outgrowth (DEVO) assay was implemented. Pooled biopsy cells were plated in a limiting dilution format at five replicates at 470,000 cells per well, 12 replicates at 100,000 cells per well, and 12 replicates at 25,000 cells per well. Cells were stimulated for 24 hr with 3 µg/mL phytohemagglutinin (PHA) (Thermofisher), 100 IU/mL IL-2, and irradiated PBMCs from an HIV-seronegative donor. The cells were washed and Molt4/CCR5 target cells were added to amplify outgrowth of HIV. The cultures were replenished with fresh media every 3–4 days, and another round of Molt4/CCR5 cells added on day 9 post-stimulation. Supernatants were harvested on days 9, 13, 16, 20, 26, and 29 post-stimulation and assessed for HIV p24 using the SIMOA HD-1 Analyzer (Quanterix, Billerica, MA). Only wells exhibiting sustained and increasing p24 expression over time were scored as positive. The frequency of infection reported as Infectious Units Per Million (IUPM) of rectal-sigmoid cells was calculated using the SLDAssay R software package (*Trumble et al., 2017*). The IUPM from this analysis was 0.265, with 95% confidence intervals between 0.0216 and 1.400. Of note, this IUPM corresponds to that of total gut cells.

## HIV DNA and transcriptional profiling analysis

Pellets of sorted cells were lysed in 1 mL TRI reagent with 2.5 µL polyacryl carrier (Molecular Research Center). Total cellular RNA was subsequently extracted per the TRI Reagent protocol, while total cellular DNA was extracted using the 'back extraction buffer' (4M guanidine thiocyanate, 50 mM sodium citrate, 1M Tris) as described (*Yukl et al., 2018*). A common reverse transcriptase (RT) reaction was used to generate cDNA for all droplet digital PCR (ddPCR) assays except TAR, where a separate two-step RT was performed as described (*Yukl et al., 2018*). Briefly, each 50 µL RT reaction contained cellular RNA, 5 µL of 10× Superscript III buffer (Invitrogen), 5 µL of 50 mM MgCl₂, 2.5 µL of 50 ng / µL random hexamers (Invitrogen), 2.5 µL of 50 µM dT15, 2.5 µL of 10 mM dNTPs, 1.25 µL of 40 U / µL RNaseOUT (Invitrogen), and 2.5 µL of 200 U / µL Superscript III RT (Invitrogen). PBMCs from uninfected donors, and water that was subjected to nucleic extraction by TRI Reagent, served as negative controls for each transcript. The thermocycling conditions were as follows: 25°C for 10 min, 50°C for 50 min, and an inactivation step at 85°C for 5 min.

For the ddPCR reactions, cDNA from each sample was assayed in duplicate wells for TAR, Long LTR, Pol, and PolyA regions using validated assays (*Kaiser et al., 2017*; *Telwatte et al., 2018*; *Yukl et al., 2018*). Total cell equivalents in the DNA extracted from the same samples were determined by measuring the absolute copy numbers of a nonduplicated cellular gene, Telomere Reverse Transcriptase (TERT), similar to previously described methods (*Telwatte et al., 2018*; *Yukl et al., 2018*). To account for the effects of deletions or hypermutations on the HIV RNA measurements, all HIV RNAs were further normalized to HIV DNA measured by the same assay used for the RNA. Each 20 µL ddPCR reaction contained 5 µL cDNA or DNA, 10 µL of ddPCR Supermix for Probes (no dUTP) (Bio-Rad, Hercules), 900 nM of primers, and 250 nM of probe. Following production of droplet emulsions using the QX100 Droplet Generator (Bio-Rad), samples were amplified under the following thermocycling conditions using a 7900 Thermal Cycler (Life Technologies): 10 min at 95°C, 45 cycles of 30 s at 95°C and 59°C for 60 s, and a final droplet cure step of 10 min at 98°C. Droplets were quantified using the QX100 Droplet Reader (Bio-Rad) and analyzed using the QuantaSoft software (version 1.6.6, Bio-Rad) in the 'Rare Event Detection' quantification mode.

## Full-length individual proviral sequencing assay (FLIPS)

The FLIPS assay was performed as previously reported (*Hiener et al., 2017*). Briefly, HIV-1 proviruses within sorted CD4+ T cell subsets were amplified and sequenced at limiting dilution to near full-length (9 kb; 92% of the genome). A median of 140 individual proviruses (range 18–254) was sequenced per sorted population of cells from three of the four participant blood specimens analyzed by PP-SLIDE. The next-generation sequencing was conducted using the Illumina MiSeq platform with individual proviruses de novo assembled using a specifically designed workflow in CLC Genomics. Proviruses were characterized as defective if they contained INDELs, stop codons, or APOBEC3G hypermutations, or intact (full-length) if they lacked such defects. Expanded identical sequences (EIS) were identified using ElimDupes from the Los Alamos database ([https://www.hiv.lanl.gov/content/sequence/elimdupesv2/elimdupes.html](https://www.hiv.lanl.gov/content/sequence/elimdupesv2/elimdupes.html)). All sequences that were 100% identical were considered part of an EIS cluster. Maximum likelihood phylogenetic trees using the generalized time-reversible model were estimated for each participant using Geneius. Branch support was inferred using 1000 bootstrap replicates. Annotated tree images were constructed using the iTOL software Version 5.5.1 (*Letunic and Bork, 2019*).

## CyTOF data export and PP-SLIDE analysis

The CyTOF data were exported as FCS files, and samples were de-barcoded according to manufacturer's instructions (Fluidigm). For the J-Lat experiments, equal numbers of the unstimulated live, singlet intact 6.3 and 5A8 were concatenated together as a source of the 'atlas cells'. Reactivated 6.3 cells were identified by gating on the Gag-expressing cells in the stimulated 6.3 sample. For patient samples, events corresponding to live, singlet intact CD3+CD19-CD8- T cells from unstimulated samples were exported as a source of the atlas cells. Events corresponding to reactivated cells were identified by gating upon live, singlet intact CD3+CD19-CD8- T cells expressing Gag on all three Gag channels, and exported as the population of reactivated cells. Where indicated, naïve cells (CD45RA+CD45RO-) were gated out before export. Data export was conducted using FlowJo (BD Biosciences) and Cytobank software. tSNE analyses were performed using the Cytobank software with default settings. All cellular markers not used in the upstream gating strategy were included in generating the tSNE plots. Non-cellular markers (e.g. live/dead stain) and the HIV gag proteins were excluded for the generation of tSNE plots. Dot plots were generated using both Cytobank and FlowJo.

PP-SLIDE analysis to identify kNN latent cells followed our previously described method (*Cavrois et al., 2017*; *Ma et al., 2020*). Using a custom script in R (source code made available as part of manuscript), each reactivated (Gag+) cell from the stimulated sample was mapped against every cell in the unstimulated atlas generated immediately after sample procurement, to identify the kNN latent cells for each patient sample.

The steps implemented for PP-SLIDE are summarized below:

### Data cleanup and standardization

The atlas cells and the reactivated cells were exported as described above. To prepare for PP-SLIDE, the following parameters, which do not contain useful information for identifying the original cell type, were removed from the analysis:

| | |
|---|---|
| Non-informative markers | Live/dead staining, event length, barcodes, beads channel, DNA, time, background channel (190), and other non-cell markers |
| Infection marker | All 3 sets of anti-Gag antibodies |
| Marker highly modified by HIV and not informative for identification of kNN latent cell | CD4 |
| Markers not expressed on CD4+ T cells | CD19, CD8, CD14 |

Raw expression values (signal intensity) of selected markers from each cell in the exported files were transformed by the inverse hyperbolic function (arcsinh) transformation as follows:

$$arsinh(x) = \ln\left( x + \sqrt{x^2 + 1} \right)$$

This transformation is used to standardize the diverse range of raw expression level scales for the measured parameters and minimizes the effect of outliers and extreme numbers.

## Identification of kNN latent cell for each reactivated cell

The Euclidean distance ($d_{F\_U}$) between each reactivated cell $F$ and each atlas cell $U$ was calculated as follows:

$$d_{F\_U} = \sqrt{\sum_{i=1}^{n}(F_i - U_i)^2}$$

where $n$ is the number of parameters analyzed and $i$ refers to the parameter being analyzed. For example, for parameter 1, $F_i - U_i$ would correspond to the value of parameter one on the reactivated cell minus the value of parameter one on the atlas cell.

For each reactivated cell $F$, the $d_{F\_U}$ of all the atlas cells $U$ were sorted from lowest to highest to identify the shortest $d_{F\_U}$ value. This corresponds to the $k = 1$ nearest neighbor atlas cell for that particular reactivated cell $F$ or the kNN latent cell. After identifying the kNN latent cell corresponding to each reactivated cell, the expression values corresponding to the original data matrix of all the kNN latent cells were combined and exported as a new FCS file for downstream analysis. These kNN latent cells correspond to a subset of the original data matrix of total atlas cells.

In the J-Lat experiments, when PP-SLIDE was implemented using all phenotyping parameters in our CyTOF panel, 6.3 cells mapped back to their parent clone with an error rate of 6.7%. Given that some parameters in our panel, such as activation markers, may not be as informative as other antigens in mapping as they markedly change upon stimulation, we undertook a systematic way to assess how the removal of the most changed parameters affects error rates. We first ranked each antigen in terms of to what extent they changed upon stimulation. As expected, the top most changed parameters were mostly activation markers. We then conducted a series of PP-SLIDE analyses where we removed the topmost changed parameter, then the top two most changed parameters, etc. As more parameters were removed, the error rate improved up to when the top six most changed antigens were removed, when the error rate reached a minimum of 0.6%. Thereafter, the error rate increased, likely because power was lost as fewer parameters were left to conduct the mapping. The top six changed parameters were those that exhibited a > 2.5 unit difference in expression levels upon reactivation. These data demonstrate that when PP-SLIDE is implemented on all phenotyping parameters it can map reactivated J-Lat 6.3 back to its correct clone with an error rate of 6.7% and that this can be improved to 0.8% when the most highly changed parameters were excluded in the analyses. Because these J-Lat PP-SLIDE results suggested that removing the most changed parameters can potentially improve the error rate, for each patient sample we conducted two sets of PP-SLIDE analyses: one using all phenotyping parameters, and one under conditions where we excluded most changed parameters (defined as a >2.5 unit difference in expression levels). The kNN latent cells identified from the two PP-SLIDE analyses exhibited similar features. The results presented correspond to those implementing all phenotyping parameters.

## Empirical cumulative distribution analysis

To assess the intra-individual and inter-individual variability in kNN latent cells and to compare kNN latent cells reactivated following treatment with PMA/ionomycin, anti-CD3/CD28, and Romidepsin/PEP005, empirical cumulative distributions were calculated. The distributions of median distances in 32-dimensional space (corresponding to measured protein levels of antigens not used in the upstream gating strategy) corresponded to those between cells of the indicated pairs of samples. The median (Euclidean) distance over all cells in the first sample was computed for each cell in the second sample using their corresponding 32-dimensional protein level vectors. Then, similarly, the median (Euclidean) distance over all cells in the second sample was computed for each cell in the first sample using their corresponding 32-dimensional protein level vectors. The union of these two sets of median distances was used to determine the distribution of median distances between the

sample pair. These resulting datasets were visualized in terms of a cumulative distribution function using the stat_ecdf function of the ggplot2 package in R.

## Distribution analysis of FlowSOM clusters

This section describes in detail how FlowSOM was implemented to cluster the atlas and kNN latent cells, and how this information was used to demonstrate non-random distribution of kNN latent cells among memory CD4+ T cells. For simplicity, we have illustrated the process with one of the four leukapheresis donors (PID1695), but similar methods were implemented for the other three. The FlowSOM analysis for all four donor specimens revealed a non-random distribution of kNN latent cells among memory CD4+ T cells.

### Clustering of atlas and kNN latent cells

The default settings of Cytobank's FlowSOM analysis pipeline (Clustering method: Hierarchical consensus; Iterations: 10; Seed: Automatic) was implemented to identify 20 clusters of memory CD4+ T cells (consists of 20 metaclusters containing 225 sub-clusters) in the atlas of unstimulated memory CD4+ T cells from PID1695 on the day of blood isolation. To characterize the latent cells, memory CD4+ T cells from the same leukapheresis were stimulated for 40 hr with PMA/ionomycin, and reactivated cells were identified based on intracellular expression of HIV-1 gag proteins. A total of 20 such reactivated cells were identified from this sample. PP-SLIDE was implemented to identify the corresponding 20 kNN latent cells from the atlas. The 20 kNN latent cells were then assigned to the appropriate cluster of the atlas. We found that the latent cells were distributed among six of the 20 clusters, while in the remaining 14 clusters no kNN latent cells were found. The distribution of all the memory CD4+ T cells is depicted as a pie graph in *Figure 2D*.

The following were the numbers of kNN latent cells ($N_{latent}$(Cluster DX) and $N_{latent}$(Cluster UX)) in each of the clusters (Clusters beginning with 'D' correspond to clusters with detectable virus, those with 'U' correspond to clusters with undetectable virus):

$N_{latent}$(Cluster D1)=3
$N_{latent}$(Cluster D2)=3
$N_{latent}$(Cluster D3)=7
$N_{latent}$(Cluster D4)=2
$N_{latent}$(Cluster D5)=4
$N_{latent}$(Cluster D6)=1
$N_{latent}$(Clusters U1-U14)=0 (undetectable)

The number of cells in each of the detectable clusters is represented as the numbers of red-colored circles in the top of *Figure 3—figure supplement 2A*.

### Proportion of latent cells in each cluster calculation

To calculate the percent of the latent reservoir in each of the detectable clusters we used the following equation:

$P_{latent}$(DX) = [(number of kNN latent cells in Cluster DX) / (total number of kNN latent cells)] * 100
This calculation gave the following percentages (P(Cluster DX)) of kNN latent cells in each detectable cluster:
$P_{latent}$(Cluster D1)=15%
$P_{latent}$(Cluster D2)=15%
$P_{latent}$(Cluster D3)=35%
$P_{latent}$(Cluster D4)=10%
$P_{latent}$(Cluster D5)=20%
$P_{latent}$(Cluster D6)=5%

The above percentages of the reservoir among the detectable clusters are depicted in blue in the top of *Figure 3—figure supplement 2A*.

## Demonstration of non-random distribution of latent cells

We then set out to determine whether kNN latent cells preferentially resided within any of the clusters, or alternatively whether the kNN latent cells were stochastically distributed. To do this, we first established the null hypothesis that kNN latent cells are randomly distributed among memory CD4+ T cells. Under this null hypothesis, the percentages of kNN latent cells in each detectable and undetectable cluster ($P_{latent\_null}$(Cluster DX) and $P_{latent\_null}$(Cluster UX), respectively) would reflect the frequency of cells making up that cluster:

$P_{latent\_null}$(Cluster D1)=3%
$P_{latent\_null}$(Cluster D2)=3%
$P_{latent\_null}$(Cluster D3)=11%
$P_{latent\_null}$(Cluster D4)=7%
$P_{latent\_null}$(Cluster D5)=15%
$P_{latent\_null}$(Cluster D6)=12%
$P_{latent\_null}$(Cluster U1)=4%
$P_{latent\_null}$(Cluster U2)=6%
$P_{latent\_null}$(Cluster U3)=3%
$P_{latent\_null}$(Cluster U4)=5%
$P_{latent\_null}$(Cluster U5)=3%
$P_{latent\_null}$(Cluster U6)=2%
$P_{latent\_null}$(Cluster U7)=2%
$P_{latent\_null}$(Cluster U8)=4%
$P_{latent\_null}$(Cluster U9)=4%
$P_{latent\_null}$(Cluster U10)=3%
$P_{latent\_null}$(Cluster U11)=8%
$P_{latent\_null}$(Cluster U12)=1%
$P_{latent\_null}$(Cluster U13)=2%
$P_{latent\_null}$(Cluster U14)=2%

We then converted these percentages into the number of latent cells we would have detected in each detectable and undetectable cluster ($N_{latent\_null}$(Cluster D) and $N_{latent\_null}$(Cluster UX), respectively) under the conditions whereby our total of 20 kNN cells were identified, which required running a total of 1,671,440 cells. This was accomplished by multiplying each percentage by this total number of kNN latent cells (20 cells), giving the following numbers of detectable kNN cells under the null hypothesis:

$N_{latent\_null}$(Cluster D1)=0.6
$N_{latent\_null}$(Cluster D2)=0.6
$N_{latent\_null}$(Cluster D3)=2.2
$N_{latent\_null}$(Cluster D4)=1.4
$N_{latent\_null}$(Cluster D5)=3
$N_{latent\_null}$(Cluster D6)=2.4
$N_{latent\_null}$(Cluster U1)=0.8
$N_{latent\_null}$(Cluster U2)=1.2
$N_{latent\_null}$(Cluster U3)=0.6
$N_{latent\_null}$(Cluster U4)=1
$N_{latent\_null}$(Cluster U5)=0.6
$N_{latent\_null}$(Cluster U6)=0.4
$N_{latent\_null}$(Cluster U7)=0.4
$N_{latent\_null}$(Cluster U8)=0.8
$N_{latent\_null}$(Cluster U9)=0.8
$N_{latent\_null}$(Cluster U10)=0.6
$N_{latent\_null}$(Cluster U11)=1.6
$N_{latent\_null}$(Cluster U12)=0.2
$N_{latent\_null}$(Cluster U13)=0.4
$N_{latent\_null}$(Cluster U14)=0.4

These numbers were then compared to the actual numbers of kNN latent cells detected in each cluster:

$N_{latent}$(Cluster D1)=3

$N_{latent}$(Cluster D2)=3
$N_{latent}$(Cluster D3)=7
$N_{latent}$(Cluster D4)=2
$N_{latent}$(Cluster D5)=4
$N_{latent}$(Cluster D6)=1
$N_{latent}$(Clusters U1-U14)=0 (undetectable)

Comparisons of $N_{latent}$(Cluster X) to $N_{latent\_null}$(Cluster X) revealed the following (where X can correspond to either detectable clusters or undetectable clusters):

Clusters D1, D2, D3, D4, and D5 were enriched for kNN latent cells relative to the null hypothesis, as defined by $N_{latent}$(Cluster X) > $N_{latent\_null}$(Cluster).

Clusters D6, U2, and U11 were disenriched for kNN cells relative to the null hypothesis, as defined by $N_{latent\_null}$(Cluster X)>1 and $N_{latent\_null}$(Cluster X)>$N_{latent}$(Cluster X). (In other words, were latent cells equivalently distributed, kNN latent cells should have been detectable in each of these clusters at frequencies higher than that actually observed.)

The remaining clusters (all undetectable ones) we could not draw conclusions on with regards to enrichment relative to the null hypothesis, as defined by the conditions $N_{latent\_null}$(Cluster X)≤1 and $N_{latent}$(Cluster X) being undetectable. (In other words, enrichment could have occurred but too few cells were run to have detected a single kNN latent cell under the null hypothesis.)

Our observation that kNN latent cells are enriched among Clusters D1-D5 and disenriched among Clusters D6, U2, and U11 refute the null hypothesis that kNN latent cells are evenly distributed among memory CD4+ T cells. These findings were further confirmed by running a chi-squared test (p<0.01).

## Frequencies of kNN latent cells among clusters

We then set out to calculate the frequencies of kNN latent cells in each of the detectable clusters. To do this, we needed to first account for the fact that more cells were analyzed in the sample harvested 40 hr post-stimulation (referred to below as 'Day 2 cells'), than in the original atlas sample. To this end, we calculated the normalization factor *NF* defined as the fold-increase in cells collected on Day 2 as compared to in the atlas:

$$NF = \text{(\# of total Day 2 cells)} / \text{(\# of total atlas cells)} = 1{,}671{,}440/11{,}974 = 139.6$$

By multiplying the numbers of cells in each detectable cluster from the atlas ($N_{day0}$(Cluster DX)) with *NF*, we obtain ($N_{atlas}$(Cluster DX)), the number of cells in each cluster we would have collected had we run by CyTOF the number of cells equivalent to that needed to detect 20 reactivated cells:

$$N_{atlas}(\text{Cluster DX}) = NF * N_{day0}(\text{Cluster DX})$$

To determine the frequencies of kNN latent cells in each cluster, we divide the number of detected kNN latent cells in each cluster ($N_{latent}$(Cluster DX)) by $N_{atlas}$(Cluster DX):

$$F_{latent}(\text{Cluster DX}) = N_{latent}(\text{Cluster DX})/N_{atlas}(\text{Cluster DX})$$

The resulting frequencies of kNN latent cells in the six detectable clusters were as follows (reported as kNN latent cells per million cells in Cluster DX):

$F_{latent}$(Cluster D1)=71
$F_{latent}$(Cluster D2)=52
$F_{latent}$(Cluster D3)=38
$F_{latent}$(Cluster D4)=17
$F_{latent}$(Cluster D5)=16
$F_{latent}$(Cluster D6)=5

These numbers are shown in the bottom of *Figure 3—figure supplement 2A*.

## Glossary of terms for stats analysis of FlowSOM clusters

| Term | Description |
|------|-------------|
| $N_{latent}$(Cluster DX) | Number of kNN latent cells in Cluster DX |
| $P_{latent}$(Cluster DX) | Percent of kNN latent cells belonging to Cluster DX |
| $P_{latent\_null}$(Cluster X) | Percent of kNN latent cells belonging to Cluster X under null hypothesis of stochastic distribution |
| $N_{latent\_null}$(Cluster X) | Number of kNN latent cells in Cluster X under null hypothesis of stochastic distribution |
| $N_{day0}$(Cluster DX) | Number of atlas cells in Cluster X that were run |
| $N_{atlas}$(Cluster DX) | Number of atlas cells in Cluster X that would have been run to capture 20 reactivated cells |
| $F_{latent}$(Cluster DX) | Frequency of kNN latent cells per million cells of Cluster DX |

## Acknowledgements

This work was supported by the National Institutes of Health (R01AI127219 and R01AI147777 to NRR; R01AI134363 to NMA; R01DK108349, R01AI132128, and R01DK120387 to SAY; UM1AI126611 to SGD; P01AI131374 to NRR and WCG), the Australian National Health and Medical Research Council (APP1149990 to SP), and the amfAR Institute for HIV Cure Research (109301). We also acknowledge NIH for the sorter (S10-RR028962), support from CFAR (P30AI027763), and the James B. Pendleton Foundation. The funders had no role in study design, data collection and analysis, decision to publish, or preparation of the manuscript. We thank M McCune for insightful discussions; N Lazarus and E Butcher for the Act1 antibody; S Tamaki, TK Peech, and C Bispo for CyTOF assistance at the Parnassus Flow Core; N Raman for assistance in flow cytometry at the Gladstone Flow Core; V Tai and M Kerbleski for assistance with the SCOPE specimens; J Carroll for assistance in graphics; F Chanut and K Claiborn for editorial assistance; and R Givens for administrative assistance.

## Additional information

### Funding

| Funder | Grant reference number | Author |
|--------|------------------------|--------|
| National Institutes of Health | R01AI127219 | Nadia R Roan |
| National Institutes of Health | R01AI147777 | Nadia R Roan |
| National Institutes of Health | R01AI134363 | Nancie M Archin |
| National Institutes of Health | UM1AI126611 | Steven G Deeks<br>Sarah Palmer |
| National Institutes of Health | P01AI131374 | Warner C Greene<br>Nadia R Roan |
| National Institutes of Health | R01DK108349 | Steven A Yukl |
| National Institutes of Health | R01AI132128 | Steven A Yukl |
| National Institutes of Health | R01DK120387 | Steven A Yukl |
| amfAR, The Foundation for AIDS Research | 109301 | Nadia R Roan<br>Warner C Greene<br>Steven G Deeks<br>Steven A Yukl<br>Peter W Hunt |
| National Health and Medical Research Council | APP1149990 | Sarah Palmer |

The funders had no role in study design, data collection and interpretation, or the decision to submit the work for publication.

## Author contributions

Jason Neidleman, Julie Frouard, Ashley Lee, Sushama Telwatte, Rebecca Hoh, Warner C Greene, Nadia R Roan, Conceptualization, Resources, Data curation, Formal analysis, Supervision, Funding acquisition, Investigation, Visualization, Methodology, Writing - original draft, Project administration, Writing - review and editing; Xiaoyu Luo, Data curation, Formal analysis, Investigation, Visualization, Methodology, Writing - review and editing; Guorui Xie, Data curation, Formal analysis, Investigation, Writing - review and editing; Feng Hsiao, Investigation, Methodology, Writing - review and editing; Tongcui Ma, Reuben Thomas, Whitney Tamaki, Formal analysis, Investigation, Methodology; Vincent Morcilla, Data curation, Formal analysis, Investigation, Methodology; Benjamin Wheeler, Investigation, Methodology; Ma Somsouk, Resources, Data curation; Poonam Vohra, Jeffrey Milush, Resources, Methodology; Katherine Sholtis James, Resources, Investigation, Methodology; Nancie M Archin, Resources, Formal analysis, Supervision, Funding acquisition, Investigation, Methodology; Peter W Hunt, Conceptualization, Resources, Formal analysis, Supervision, Funding acquisition, Methodology; Steven G Deeks, Conceptualization, Resources, Funding acquisition, Writing - review and editing; Steven A Yukl, Resources, Supervision, Funding acquisition, Writing - review and editing; Sarah Palmer, Resources, Formal analysis, Supervision, Funding acquisition, Methodology

## Author ORCIDs

Vincent Morcilla http://orcid.org/0000-0002-8159-6090
Benjamin Wheeler http://orcid.org/0000-0001-5310-7213
Steven G Deeks https://orcid.org/0000-0001-6371-747X
Nadia R Roan https://orcid.org/0000-0002-5464-1976

## Ethics

Human subjects: The study was approved by the University of California, San Francisco (IRB # 10-01330) and the University of North Carolina (IRB # 12-1660). All participants provided informed consent before participation.

## Decision letter and Author response

Decision letter https://doi.org/10.7554/eLife.60933.sa1
Author response https://doi.org/10.7554/eLife.60933.sa2

# Additional files

## Supplementary files

- Source code 1. Source code for PP-SLIDE identification of kNN latent cells.

- Supplementary file 1. List of CyTOF antibodies used in the study. Antibodies were either purchased from the indicated vendor or prepared in-house using commercially available MaxPAR conjugation kits per manufacturer's instructions (Fluidigm).

- Supplementary file 2. Participant characteristics. Table of participant characteristics listing gender, ethnicity, age, year of first HIV+ test, viral load, CD4 count at the time of sampling, ART regimen, and specimen type used in this study.

- Transparent reporting form

## Data availability

Raw CyTOF datasets have been made publically available through the public repository Dryad: https://doi.org/10.7272/Q6KK991S. The following is the citation for this dataset: Neidleman et al. (2020), Phenotypic Analysis of the Unstimulated In Vivo HIV CD4 T Cell Reservoir, v2, UC San Francisco, dataset, https://doi.org/10.7272/Q6KK991S.

The following dataset was generated:

| Author(s) | Year | Dataset title | Dataset URL | Database and Identifier |
|---|---|---|---|---|
| Neidleman J, Luo X, Frouard J, Xie G, Hsiao F, Ma T, Morcilla V, Lee A, Telwatte S, Thomas R, Tamaki W, Wheeler B, Hoh R, Somsouk M, Vohra P, Milush J, James K, Archin NM, Hunt PW, Deeks SG, Yukl SA, Palmer S, Greene WC, Roan NR | 2020 | Phenotypic Analysis of the Unstimulated In Vivo HIV CD4 T Cell Reservoir | https://doi.org/10.7272/Q6KK991S | Dryad Digital Repository, 10.7272/Q6KK991S |

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
