## [Decision Letter]

**Acceptance summary:**

This manuscript describes a method that leverages validated high-dimensional phenotyping to trace latently HIV-infected cells. The results suggest that contrary to common assumptions, the reservoir is not randomly distributed among cell subsets, and is conserved between individuals. This work addresses an important area of research as the identification and characterization of the latent HIV reservoir is critical to efforts to develop curative strategies. The techniques described provide a tool for identifying the latently-infected CD4+ T cell subsets and suggests potential applications in the design of therapeutic strategies to precisely study and target latently-infected cells.

**Decision letter after peer review:**

Thank you for submitting your article "The Atlas of the in vivo HIV CD4 T Cell Reservoir" for consideration by *eLife*. Your article has been reviewed by three peer reviewers, and the evaluation has been overseen by a Reviewing Editor and Satyajit Rath as the Senior Editor. The following individuals involved in review of your submission have agreed to reveal their identity: Catherine Blish (Reviewer #1); Sara Gianella (Reviewer #3).

The reviewers have discussed the reviews with one another and the Reviewing Editor has drafted this decision to help you prepare a revised submission.

Summary:

This well-written manuscript describes results from combining single-cell phenotyping with PP-SLIDE analysis to map HIV latently-infected CD4+ T cells that are activated ex vivo to their original status in the blood, lymph nodes and gut of the patients who have been taking ART. This work addresses an important area of research as the identification and characterization of the latent HIV reservoir is critical to efforts to develop curative strategies. The techniques described provide a tool for identifying the latently-infected CD4+ T cell subsets and suggests potential applications in the design of therapeutic strategies to precisely study and target latently-infected cells.

Essential revisions:

A) The term Atlas is too strong given the small numbers studied and should be changed to something less 'global'.

Additionally, there are three major issues:

1) One is that the sample sizes are too small to draw the broad conclusions you have made both between individuals and within an individual. Ideally, a larger sample size would make this story stronger. However, the reviewers did appreciate the novelty of even this more limited study and therefore if it is not possible to test more samples, require that you really tone down the conclusions to reflect the sparse sampling. For this to be an "atlas" they expected that a deeper phenotyping such as that proposed with the single-cell RNA-sequencing, and a greater number of patients with a broader range of LRAs tested. So, a less global word should be used in the title.

2) Related to the first point, concerns were also raised about the claims of similarity based on Euclidean distance, which simply cannot be supported by the data, as well as the claims of repertoire stability when each comparison was only 1 pair. The reviewer noted that in Figure 4E, it's hard to say, as it is very difficult to tell the difference between the colors for the gut vs. FNA. In fact, it looks like they are forming distinct clusters, but that a single different cell in the FNA is driving the overlap between the Gut and FNA samples. If I'm interpreting it correctly, 6 FNA cells make a distinct cluster at the bottom with only 1 at the top. However, the Euclidean distance would average that out, calling into question the validity of the finding that the median and mean Euclidean distances, as one distinct cell seems to be driving the entire overlap. The conclusions may therefore need to be tempered or additional samples added to resolve this.

3) There were concerns on the staining and time of incubation. They require you to show the gag staining and the degree of enrichment using your combination of markers. For the staining, you should show the p24 staining used in the Cytof to classify the reactivated cells for the 4 donors. For the memory markers, CD27 and CCR7 should be bimodal, thus, it is important to explain how you classify the different memory subset CM, TM and EM and get distributions that are very different to what is commonly observed in people. You should also address the way you selected the disenriched population to compare to their enriched population as some of them like naive cells (CD62L+ CD27+) have low proviral content or are at extremely low frequency compared to the enriched population therefore having less proviruses as well.

---

## [Author Response]

Essential revisions:A) The term Atlas is too strong given the small numbers studied and should be changed to something less 'global'.

We have modified the title to not use the term “atlas,” and have removed “atlas” from the list of key words. The only remaining use of the term “atlas” is when we refer to the total population of unstimulated memory CD4+ T cells from each patient sample, and we have defined this use of the word in this context. As this use of the term “atlas” is now technical, simply to have a one-word phrase that is easy to refer to (instead of repeatedly stating “unstimulated memory CD4+ T cells from the same subject” which is unnecessarily wordy), we believe it appropriate to use the term in this context. Of note, the paper no longer states our having established an “atlas” of the HIV reservoir.

Additionally, there are three major issues:1) One is that the sample sizes are too small to draw the broad conclusions you have made both between individuals and within an individual. Ideally, a larger sample size would make this story stronger. However, the reviewers did appreciate the novelty of even this more limited study and therefore if it is not possible to test more samples, require that you really tone down the conclusions to reflect the sparse sampling. For this to be an "atlas" they expected that a deeper phenotyping such as that proposed with the single-cell RNA-sequencing, and a greater number of patients with a broader range of LRAs tested. So, a less global word should be used in the title.

We have now acknowledged within the Discussion the small sample size as a limitation of the study. That being said, we would like to point out that the findings we reported in the 4 blood donors analyzed by PP-SLIDE were validated in four out of four new and completely untested donors (Figure 6F, previously named Figure 5F), demonstrating that the broad conclusions we had made from the original (limited number of) 4 donors can be generalizable. Although it is possible that if we test yet another four donors we may see some outliers (something that is not possible to perform as recruiting additional HIV-infected participants for this study are on hold due to COVID-19), we think our validation studies allow us to be somewhat confident in our conclusions. Furthermore, as noted above, we have now removed any use of the term “atlas” when referring to our characterization of the HIV reservoir.

2) Related to the first point, concerns were also raised about the claims of similarity based on Euclidean distance, which simply cannot be supported by the data, as well as the claims of repertoire stability when each comparison was only 1 pair. The reviewer noted that in Figure 4E, it's hard to say, as it is very difficult to tell the difference between the colors for the gut vs. FNA. In fact, it looks like they are forming distinct clusters, but that a single different cell in the FNA is driving the overlap between the Gut and FNA samples. If I'm interpreting it correctly, 6 FNA cells make a distinct cluster at the bottom with only 1 at the top. However, the Euclidean distance would average that out, calling into question the validity of the finding that the median and mean Euclidean distances, as one distinct cell seems to be driving the entire overlap. The conclusions may therefore need to be tempered or additional samples added to resolve this.

We apologize for the similar colors between the FNA and gut, we had thought to use two shades of blue to have tissue specimens all be colored blue. We have now changed the figure such that the FNA and gut specimens are completely different colors.

With regards to the validity of Euclidean distance measurement calculations, we would like to point out that we did not average any of the Euclidean distances. In the example the reviewer gave, he/she commented that one single outlier cell could have driven an inaccurate conclusion of overall similarity between two populations. In fact, we had used median (not mean) measurements to make sure that such outliers would not skew our data. In addition to showing the density plots, we also showed the data as empirical cumulative distribution curves, where the y-axis value corresponds to the proportion of the median distances that are less than the corresponding value indicated on the x-axis. These plots are the most quantitative and rigorous way of demonstrating the relative similarities of populations that are being compared.

The reviewer had specifically commented that a single cell in the FNA sample may have biased our results, as it appeared far away from another 6 cells in the tSNE plot. We would first like to point out that there were in fact a total of 49 FNA cells shown in the tSNE plot (not readily apparent because some cells were stacked on top of one another, but in fact they consist of the sum of the FNA kNN latent cells shown in Figure 4C), and that the “single” cell that the reviewer referred to is actually two cells stacked on top of one other. More importantly, tSNE is simply a visualization tool, and should not be used to draw conclusions about how similar populations are to one another. Furthermore, it is a visualization tool that is good at depicting local population relationships, but does not accurately depict long-distance relationships. To draw quantitative and statistically-sound conclusions about how similar or different populations to one another, other computational tools not dependent on tSNE, such as the Euclidean distance calculations we had implemented, are required.

To prove our point that the 2 FNA cells towards the top of the tSNE are not artificially driving our conclusions as the reviewer suggested, we, as an exercise, removed those two cells from the datasets:

**Author response image 1. sa2fig1:** 

We then repeated the Euclidean distance calculation analysis with the remaining 47 cells FNA cells, and the original numbers of blood and gut cells. Even when the two FNA cells at the top of the tSNE are removed, it was clear that the FNA specimens are more similar to the gut than to the blood specimens (see Figure 4E).We therefore strongly believe our Euclidean distance calculation algorithm is a valid method for quantifying similarities/differences between populations and that the original conclusions we had drawn are sound.

3) There were concerns on the staining and time of incubation. They require you to show the gag staining and the degree of enrichment using your combination of markers. For the staining, you should show the p24 staining used in the Cytof to classify the reactivated cells for the 4 donors.

The staining protocol we had implemented was identical to that which we have previously used and validated to examine in vitro infection with HIV (Ma et al., 2020) and which was referenced in the Materials and methods section. The p24 staining (using two different anti-p24 antibodies) that was used in the CyTOF to classify reactivated cells for the 4 donors was shown Figure 2B (Of note, we referred to p24 as “Gag” since technically speaking, intracellular Gag is p55, not p24).

In addition to showing the staining, we have also indicated, for each donor, the numbers of reactivated cells identified by this staining p24.

In the original Figure 5—figure supplement 1 (now Figure 5, since a reviewer suggested to convert this original supplemental figure to a main one), we had shown the fold-enhancement of kNN latent cells in the final enriched population relative to each of the disenriched ones. This was calculated by dividing the frequency of kNN latent cells in the enriched population by the frequency of kNN latent cells in each of the disenriched populations. We now show within this figure the original frequency of kNN latent cells for each of the enriched populations, as requested. We have also added the kNN frequency information to the original Figure 5—figure supplement 5 (now renamed Figure 6—figure supplement 5), where shared features between kNN latent cells of different donors were used to design a universal sort panel.

For the memory markers, CD27 and CCR7 should be bimodal, thus, it is important to explain how you classify the different memory subset CM, TM and EM and get distributions that are very different to what is commonly observed in people.

We have now added to Figure 3—figure supplement 1 our strategy for identification of the Tcm, Ttm, and Ttm populations based on expression levels of CD27 and CCR7.

Of note, while CD27 staining on T cells is typically bi-modal, CCR7 is not. This is true not only in CyTOF data (this study, Ma et al. eLIFE, 2020, Neidleman et al. Cell Reports Medicine 2020, Cavrois et al. Cell Reports, 2017), but also in FACS data. As an example of such FACS data, we have shown in Author response image 2 a figure from a paper (Soriano-Sarabia et al., 2014) that used flow cytometry to sort CD4+ Tcm and Ttm from ART-suppressed HIV-infected individuals to demonstrate that the replication-competent reservoir is more enriched in Tcm than Ttm. Similar to what we have seen by CyTOF, CCR7 (here on the y-axis) stains as one continuous smear.

We also note that the frequencies of Tcm, Ttm, and Tem that we report, with Tcm being the most abundant and Ttm the least, are similar to those reported by this prior and other studies.

You should also address the way you selected the disenriched population to compare to their enriched population as some of them like naive cells (CD62L+ CD27+) have low proviral content or are at extremely low frequency compared to the enriched population therefore having less proviruses as well.

It is precisely because naïve cells from blood have low proviral content that we did not want to include them within the comparison group. In fact, we feel that one rather concerning aspect is that many published studies in the field reporting surface markers preferentially expressed on HIV reservoir cells are simply markers that are preferentially expressed on memory cells and not naïve cells. If a marker X is preferentially expressed on memory cells and not naïve cells, then sorting CD4+ T cells expressing marker X will enrich for the reservoir simply by enriching for memory cells. For this reason, we also believe it is very important to compare any potential “enriched” population of cells to a memory T cell population, not to total T cells.

Importantly, *none* of our disenriched populations had any naïve cells; that in our view would not be a fair comparison. In fact, we went to great lengths to choose populations of “disenriched” cells that – prior to our study – the field would believe to be very much enriched for the reservoir. All of our disenriched populations were memory CD4+ T cells expressing PD1. We chose to focus on PD1 because it is a surface protein that is well-accepted to be more highly expressed in HIV reservoir cells, and was demonstrated to enrich for replication-competent HIV (unlike other surface proteins which were only shown to enrich for total HIV DNA, a concern as most of the in vivo reservoir consists of defective proviruses). In essence, we had set a high bar for comparing our enriched populations to PD1-expressing memory CD4+ T cells, but did so because we wanted to prove that PP-SLIDE is a robust platform that enables us to enrich already-enriched populations of cells. This proved to be the case as in all 8 donors in which we conducted viral outgrowth assays, our enriched populations harbored more replication-competent HIV than the corresponding PD1+ disenriched populations. With regards to how the markers other than PD1 were chosen in the disenriched populations, this was through a combination of manually assessing each surface markers’ expression levels on the kNN cells, the numbers of cryopreserved cells available from the participant, and the overall abundance of the populations of cells in each sequential gate, as stated in the subsection “Markers of kNN latent cells enrich for the replication-competent, genome-intact reservoir”.

These important aspects describing the rationale for comparing our enriched populations to disenriched populations of PD1+ memory cells are discussed in the third paragraph of the Discussion.